# A Survey of Information Extraction Based on Deep Learning

**Yang Yang, Zhilei Wu, Yuexiang Yang \*, Shuangshuang Lian, Fengjie Guo and Zhiwei Wang**

School of Management, China University of Mining and Technology (Beijing), Beijing 100083, China
\* Correspondence: 201901@cumtb.edu.cn

**Abstract:** As a core task and an important link in the fields of natural language understanding and information retrieval, information extraction (IE) can structure and semanticize unstructured multi-modal information. In recent years, deep learning (DL) has attracted considerable research attention to IE tasks. Deep learning-based entity relation extraction techniques have gradually surpassed traditional feature- and kernel-function-based methods in terms of the depth of feature extraction and model accuracy. In this paper, we explain the basic concepts of IE and DL, primarily expounding on the research progress and achievements of DL technologies in the field of IE. At the level of IE tasks, it is expounded from entity relationship extraction, event extraction, and multi-modal information extraction three aspects, and creates a comparative analysis of various extraction techniques. We also summarize the prospects and development trends in DL in the field of IE as well as difficulties requiring further study. It is believed that research can be carried out in the direction of multi-model and multi-task joint extraction, information extraction based on knowledge enhancement, and information fusion based on multi-modal at the method level. At the model level, further research should be carried out in the aspects of strengthening theoretical research, model lightweight, and improving model generalization ability.

**Keywords:** deep learning; information extraction; entity relationship extraction; event extraction; multi-modal information extraction

## 1. Introduction

With the rapid development of the Internet, information resources are extremely abundant, but the problem of "information overload" is becoming increasingly serious. It is urgent to obtain information quickly and accurately. Information extraction technology came into being. Information extraction is a text processing technology that extracts the specified type of entity, relationship, event, and other factual information from natural language text and forms a structured data output. The goal of information extraction is to make the information machine readable, which is the foundation and core of natural language processing. Although automatic information retrieval has been a mature subject, its history is as long as that of document databases, however, automatic information extraction technology has been developed in the past decade. The development history of information extraction technology is similar to the development history of artificial intelligence. It has undergone three iterations: the method based on rules and dictionaries, the method based on statistical machine learning, and the method based on deep learning, as shown in Table 1.

In recent years, DL has become the mainstream technology in the field of natural language information extraction because of its strong feature extraction and learning ability. The concept of deep learning was first proposed by Hinton in 2006 to study how to automatically extract multi-layer feature representations from data. Its core idea is to extract features from the original data from the low-level to the high-level and from the concrete to the abstract through a series of non-linear transformations in a data-driven manner [1]. Deep learning allows a computing model composed of multiple processing layers to learn multi-level abstract data representations. These methods have greatly improved the

advanced levels of speech recognition, visual object recognition, object detection, and many other fields [2]. Different from traditional shallow learning, deep learning emphasizes the depth of the model structure, and obtains deep meaning by increasing the model's depth. Secondly, deep learning clarifies the importance of feature learning, and transforms the feature representation of the sample in the original space into a new feature space through layer-by-layer feature transformation, thus, making classification or prediction easier. Compared with classical machine learning, the characteristics of DL are shown in Table 2.

**Table 1.** The Development history of IE.

| IE Methods | Main Ideas | Characteristics |
|---|---|---|
| Methods based on rule and dictionary | Rule based methods: by summarizing the rules, experts construct a large number of rule templates, and extract information based on the templates. Dictionary based methods: to establish a dictionary of recognition objects; the process of information extraction is the process of searching in the dictionary or professional domain knowledge database. | The method based on manual rules can achieve high accuracy on small data sets, but it has no adaptability to a large number of data sets or new fields. The establishment of new rule bases and dictionaries requires a lot of time and manpower. These rules often depend on specific languages, and it is difficult to cover all languages. |
| Methods based on statistical machine learning | Supervised training is carried out by using the manually labelled corpus, and then the prediction is realized by using the trained machine learning model. Common methods include HMM, MEM, SVM, ME, CRF, etc. | Although the method based on statistical machine learning has significantly improved results compared with the previous methods, it also requires a lot of manual annotation by people with professional field knowledge, and the cost of labor and time is very high. |
| Methods based on deep learning | The complex pattern recognition problems are solved by automatically identifying information features and internal laws through complex network structures. Common methods include CNN, RNN, etc. | It is applicable of big data processing and automatically learns sentence features without complex feature engineering. |

Currently, the rapid development of deep learning has attracted widespread attention from academia and industry. Owing to their excellent feature selection and extraction abilities, these technologies have exerted an increasingly important influence on many tasks, including machine translation, object recognition, and image segmentation. Meanwhile, natural language processing (NLP), computer vision (CV), and speech recognition (SR) have been widely implemented.

Therefore, the use of deep learning technology to promote the development of the field of natural language processing is the focus of current research. At the same time, scholars at home and abroad have devoted considerable effort and attention to this field and have carried out a great deal of research. Although existing deep learning algorithm models, such as CNNs and RNNs, have been widely used in the field of natural language processing, no major breakthroughs have been recently reported. It may be considered that research on deep learning in the field of natural language processing is still in its infancy, and a series of problems remain to be solved around DL-IE.

**Table 2.** Differences between deep learning and classical machine learning.

| Characteristic | Deep Learning | Classical Machine Learning |
|---|---|---|
| Data requirements | It is suitable for processing big data. With the increase in data volume, its performance will also be improved. Its dependence on prior knowledge of the data is weak. | It is suitable for small- and medium-sized data and has a strong dependence on prior knowledge of the data. |
| Model structure | The model has high complexity, a wide application range, and good expansibility. | It has a simple network structure with poor portability. |
| Feature extraction | Without feature engineering, it can automatically learn feature representations, which makes it easier to find hidden features and improves generalization ability [3]. | Features need to be identified by experts, and then manually coded according to the domain and data type. There are problems on error accumulation and propagation. |
| Solution | Solves problems once and end-to-end, with strong adaptability and easy migration. | Solves the problem in stages and then re-assemble. |
| Execution time | It takes a lot of time to train and there are too many parameters to learn. | Generally, it can be trained well in a few seconds to a few hours. |
| Interpretability | Due to the lack of theoretical basis, the deep-seated network cannot be explained, and the hyperparameters and network design are also a great challenge. | Rules and characteristics are understandable. |
| Hardware dependency | Deep learning algorithm has high requirements on GPU and relies on high-end hardware facilities. | It can work on low-end machines. |

In a review of the existing relevant literature, Xu et al. discussed the progress of deep learning in terms of word annotation, syntactic analysis, emotion analysis, machine translation, and text classification [4]. He believes that the accuracy of the results of deep learning algorithms depends on the amount of training data. Combining existing knowledge (i.e., traditional machine learning methods) with deep learning methods to improve learning efficiency is the next research direction. Luo introduced convolutional neural networks and recurrent neural networks and detailed the BERT, XLNet, and ERNIE3 pretraining language models, and argued that many challenges remain to be overcome for deep learning in NLP to be implemented on a large scale, despite its dramatic recent great successes [5]. Yangyang et al. used CiteSpace and VOSviewer to draw a knowledge graph of the research countries, institutions, journal distribution, keyword co-occurrence, cocited network clustering, and time axis view of deep learning in the field of natural language processing to clarify the research context. By reviewing the important literature in the field, the research trends, main problems, and development bottlenecks of deep learning in the field of natural language processing were summarized, and the corresponding solutions and ideas were provided [6]. Liu et al. divided the common problems in the implementation of multi-modal deep learning into four categories: modal representation, modal interpretation, modal fusion, and modal alignment, and sub-classified and discussed various problems [7].

It can be seen that most of the previous literature reviews analyzed the pre-training model, field application, research trajectory, and distribution of NLP. There are few in-depth analyses and elaborations from the perspective of IE, the foundation and core issue of deep learning in natural language processing. Therefore, we review the latest literature on the

subject published in recent years, focusing on the practice of DL in IE from the perspectives of entity relationship extraction, event extraction, and multi-modal IE as three aspects of research progress analysis. The research framework is shown in Figure 1. Among them, the characteristics and applications of typical models are shown in Appendix A. We expect this work to provide a reference to facilitate subsequent research and development.

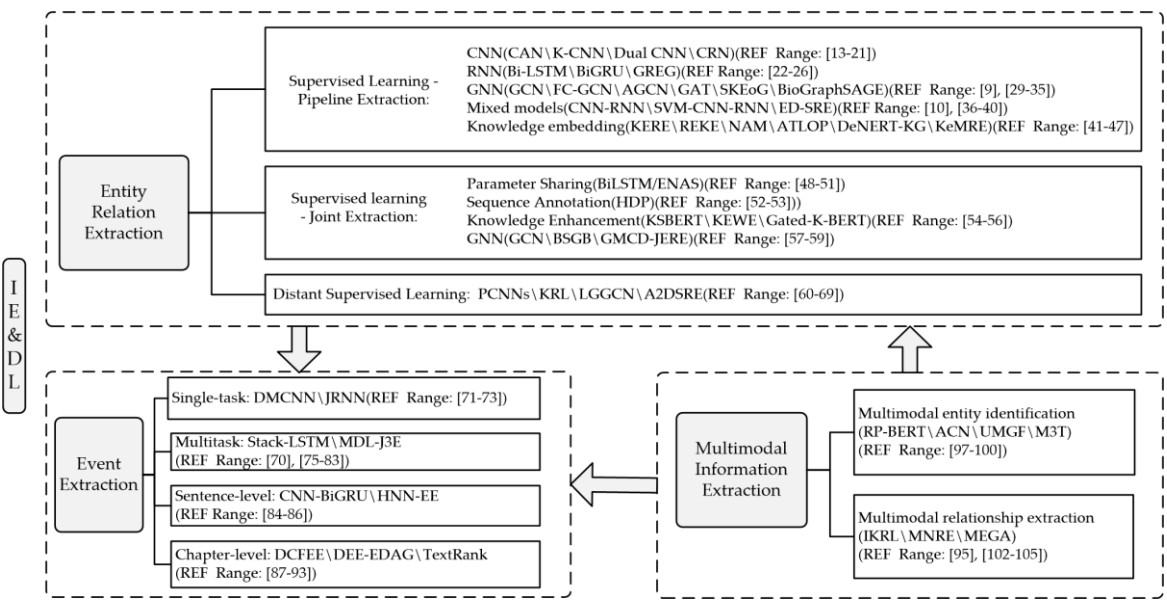

**Figure 1.** Overall architecture of IE based on DL.

## 2. Entity Relationship Extraction Based on Deep Learning

Entity relationship extraction is a basic task in the field of NLP. The task is defined as specifying the appropriate relationship types for entity pairs in sentences. Accurate relationship extraction results facilitate accurate text interpretation, discourse processing, and higher-level natural language processing systems. Various relationships extracted from text contribute to the construction of knowledge graphs and downstream tasks that require a relational understanding of text, such as intelligent question answering, bio-medical knowledge discovery, and dialog systems.

Compared with traditional methods, the advantage of deep learning techniques lies in the ability to learn an appropriate feature representation for the current problem in an end-to-end manner, replacing the hand-crafted features of the past [8]. Early rule-based relation extraction methods manually constructed language rules to represent the semantic features of sentences, realizing entity recognition and relation-type discrimination. Rule-based entity relationship extraction can be divided into two categories, including trigger word- and dependency-based relationship extraction methods. Manual construction rules can be tailored for specific fields with high accuracy and are easy to implement on small-scale datasets. However, based on the rules of different relationship extraction methods, which depend on domain knowledge and artificial feature extraction, the specific rules of different fields require experts to construct the learning models. Considering all possible rules is difficult and requires considerable time and energy. The performance of relationship extraction methods depends on the quality and scale of the rules, and exhibits

the characteristics of poor field-migration performance and low recall. Subsequently, a machine learning-based relationship extraction method was proposed. Machine learning-based relationship extraction methods use feature engineering and annotation data to obtain better performance, effectively reducing the dependence on linguistics and domain knowledge, and have shown strong domain transfer abilities. However, eigenvector- and kernel-function-based methods can produce error propagation through learning pipelines, which greatly limits the performance of the model [9].

By avoiding tedious manual feature extraction and improving error propagation in feature extraction, relationship extraction methods based on deep learning have become a focus of recent research [10]. Compared with traditional machine learning based on statistical models, deep learning-based methods use word vectors in text as input to achieve end-to-end feature extraction through neural networks, and no longer rely on manually defined features [11]. At present, entity relationship extraction based on DL can be roughly divided into supervised and distant supervised learning-based relationship extractions according to the training methods, in which relationship extraction based on supervised learning includes pipeline extraction and joint extraction.

### 2.1. Supervised Learning—Pipeline Extraction

The pipeline extraction method refers primarily to the extraction of entity recognition, followed by relationship extraction. The early pipeline extraction learning methods included CNN and RNN models. The properties of CNN diversity convolutional kernels are beneficial for identifying the structural features of a target [12]. The architecture of CNN models is relatively simple, mainly including a convolutional layer in the front and a fully connected layer behind, and they can be trained relatively quickly. RNNs are neural networks used to process sequence data. RNNs have both internal feedback connections and feedforward connections between the processing units and can use their internal memory to process the sequence information of arbitrary timing, with the ability to learn the combined vector representations of various phrases and sentences of arbitrary length. Compared to general neural networks, they can handle sequence-changing data. However, RNNs also have some limitations, such as gradient disappearance and explosion, as well as long network training cycles; therefore, traditional RNNs struggle to handle long-term dependence in practice. With the continuous development of deep learning, relationship extraction methods based on CNNs and RNNs have been studied, and many variants have been produced. These include dual convolutional neural network (dual CNN), convolutional attention network (CAN), long short-term memory (LSTM), and bi-directional long short-term memory network (Bi-LSTM) models. Additionally, there are variants of graph neural networks (GNNs), graph convolutional networks (GCNs), and graph attention networks (GATs), such as graph convolutional neural network-based feature combinations (FC-GCNs) and graph convolutional network-based attention (AGCN), which are also being gradually used for entity relationship extraction. In addition, researchers are also committed to developing the optimal combination of different models and extraction experiments using existing knowledge embeddings to continuously improve the model effect.

#### 2.1.1. CNNs

Zeng et al. applied deep convolutional neural networks to perform relationship extraction, which is feature extraction by a neural network that avoids manual feature extraction and realizes end-to-end relationship extraction [13]. Solid relation extraction using a simple CNN model consisting of input, convolution, pooling, and softmax layers was performed by Liu et al. [14]. In order to reduce the application limitations of CNNs, Lavin et al. introduced a fast algorithm for CNNs based on Winograd's minimum filtering algorithm to improve the implementation efficiency of the convolutional neural network algorithm [15]. Gu et al. conducted an in-depth analysis of the CNN and considered that CNN has good prospects in the application of images, videos, voice, and text recognition, but the CNN

algorithm still needs to be optimized. At the same time, they discussed the improvement of CNN in the aspects of layer design, activation function, loss function, regularization, optimization, and fast calculation. Gu also proposed a language CNN model for a statistical language modeling task [16,17]. Zhou et al. proposed a new convolution attention network (CAN) for chemical-disease relationship (CDR) extraction, which is designed to perform convolution operations on the shortest dependency path (SDP) between chemical and disease pairs to produce deep semantic dependence features. An attention mechanism is then employed to capture weighted semantic-dependent representations related to knowledge representations learned from the knowledge database. It mainly consists of a representation layer, a convolutional layer, a knowledge-based attention layer, and a softmax layer. The experimental results show that dependent information and prior knowledge were effective for CDR extraction and that prior knowledge was able to significantly improve performance, but the method only considered intra- and inter-sentence CDR extraction and ignored the uniformity of documents [18].

Li et al. proposed a knowledge-oriented convolutional neural network (K-CNN) designed to perform causality extraction. The K-CNN architecture contains two collaborative channels (as shown in Figure 2), including knowledge-oriented channels and data-oriented channels. These models combine both prior human knowledge and the information derived from data in a complementary manner to extract causality from natural language texts. Filter selection and clustering techniques have also been proposed to reduce dimensionality and improve the performance of the K-CNN. To the best of our knowledge, the automatic selection of target entities identified by causality as well as the more efficient extraction of complex causality has not yet been investigated in the relevant literature. Future attempts can consider the application of K-CNN models to relationship extraction tasks other than causality and explore potential applications of K-CNN in other domain-specific tasks [19]. A dual convolutional neural network (dual CNN) model based on a knowledge-attention mechanism was proposed by Li et al. The model includes an input layer, a convolutional layer, a pooling layer, a merging layer, and a fully connected layer (as shown in Figure 3). The model inserts word embeddings and supervised information from the knowledge database into the CNN, performs convolutions and pools, and combines a knowledge database and a CNN structure in the fully connected layer [20]. Yu et al. proposed a relationship extraction method for domain knowledge graph construction. According to the structure and content characteristics of the knowledge in the network encyclopedia, the relationship extraction task was divided into upper and lower relationship extraction and non-superordinate relationship extraction, using co-occurrence analysis and semantic analysis to extract the upper and lower relationships of classification labels and through the improved convolution residual network (CRN) of the cross entropy loss function to extract non-superordinate relations from the unstructured text [21].

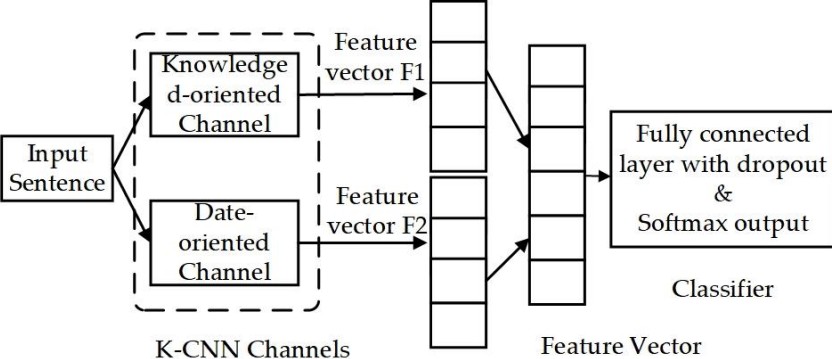

**Figure 2.** The overall architecture of K-CNN, which contains two collaborative channels, including knowledge-oriented channels and data-oriented channels, combines both prior human knowledge and the information derived from data in a complementary manner to extract causality from natural language texts [19].

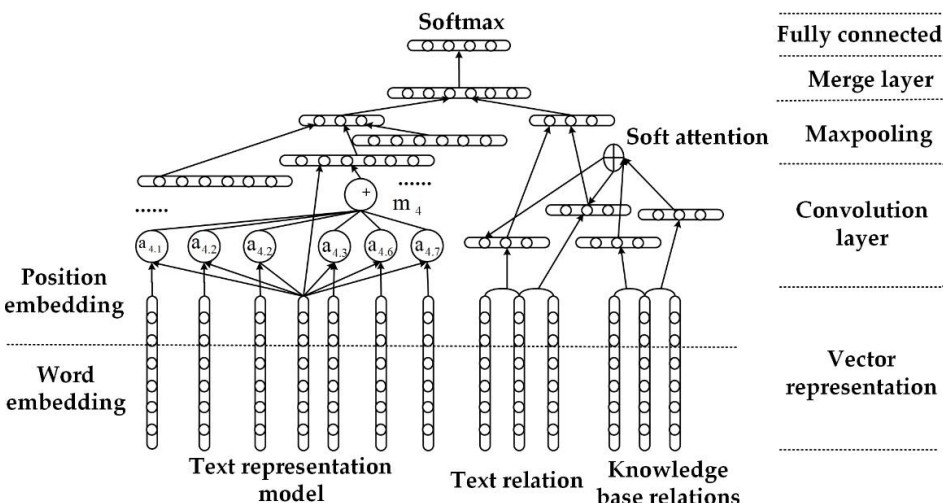

**Figure 3.** The architecture of the dual CNN, includes an input layer, a convolutional layer, a pooling layer, a merging layer, and a fully connected layer [20].

### 2.1.2. RNNs

Compared with statistics-based methods, CNN-based methods have shown good progress, but their ability to extract temporal features is weak, especially when the "distance" between two entities is relatively large, which limits their performance. Circulating neural networks are considered the most suitable model for temporal feature extraction. In network training, a traditional RNN is prone to gradient disappearance and gradient explosion problems; therefore, RNNs have difficulty dealing with long-term dependence in practice [22]. The LSTM architecture is a special RNN that solves this problem. LSTM models perform better for longer sequences than ordinary RNNs. To improve the performance of these models in handling multiple entities in long text and sentences, Li et al. proposed a relationship-classified neural network structure based on bi-directional long- short-term memory networks. The model implemented a concat-attention mechanism to capture the most important context words in sentences, a piecewise attention mechanism to improve its performance in processing long sentences, and a tensor-based entity description to overcome the problem of performance declines when multiple entities appear in a sentence [23]. A recurrent neural network with multiple semantic heterogeneous embeddings was built in a self-training framework, as proposed by Lin et al. The framework uses labelled, unlabelled, and social media data to model relational context using a bi-directional recurrent neural network with scalability and versatility. Compared with the SVM model utilizing complex features, the RNN-based time-relation extracted self-training framework performed well on the original features, modelled the sentence structure well, and was highly scalable and generalizable [24].

At present, most relationship extraction methods extract the relationships reflected by a single entity pair in a certain sentence; however, in long text, prior methods focused on the representation of entire sentences while ignoring the loss of information. The lack of analysis of words and syntactic characteristics leads to poor sentence performance, especially in Chinese relationship extraction. Zhang et al. constructed a bi-directional gate recursion unit (BiGRU) network relationship extraction model based on character- and sentence-level attention mechanisms to extract the relationship between diseases, symptoms, and tests [25]. An end-to-end relationship extraction method based on a bi-directional gated recursive unit (BiGRU) neural network and a dual attention mechanism was proposed by Yue et al. The model is designed to focus on words with a decisive influence on sentence-relation extraction and capture relational semantic and directional words. To improve performance, it generates word vectors using a pre-trained FastText model and dynamically adjusts word vectors via FastText according to the context [11]. A large number of relational facts are expressed in multiple sentences. Complex inter-

relationships often exist between multiple entities in a document. Some relationships can become coherent only on a global scale. Kim et al. proposed a global-level relationship extractor model, GREG (as shown in Figure 4), which contained two modules, including a context-aware long-and short-term memory (LSTM) relationship extraction module and a knowledge graph constructor module for generating a knowledge graph from a given document. An unsupervised version of this method is planned in the future, which would be much closer to full automation [26].

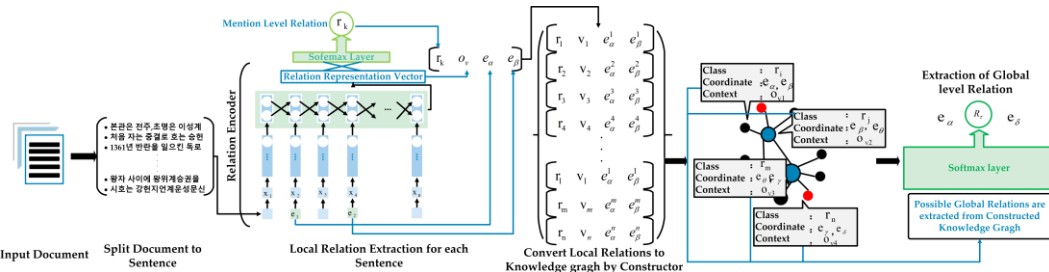

**Figure 4.** Global-level relationship extractor model GREG, which contained two modules, including a context-aware LSTM relationship extraction module and a knowledge graph constructor module for generating a knowledge graph from a given document [26].

### 2.1.3. GNNs

RNN models can only deal with directed positional acyclic graphs (DPAGs), and are generally used to deal with graph-focused problems. Graph neural networks (GNNs) process input data encoded as generic labeled graphs. Graph neural networks can model node-centric and graph-centric functions. GNN can be used for sentence extraction, document clustering, and classification [27]. CNN can only process regular Euclidean data, such as images (two-dimensional grid) and text (one-dimensional sequence), for non-Euclidean data, the processing results are not satisfactory. The disadvantage of the graph embedding model is that the parameters are not shared among the nodes in the encoder and the number of parameters grows linearly with the number of nodes, resulting in low computational efficiency. Second, the direct embedding method lacks the ability to generalize to dynamic graphs and cannot be generalized to new graphs. Based on CNN and graph embeddings, a graph neural network (GNN) is proposed to collectively aggregate the information in the graph structure. Models of GNN can be divided into graph convolutional networks (GCN), graph attention networks (GAT), graph spatial-temporal networks (GAT), etc. [28]. Currently, researchers more often use the GCN model and GAT model for relationship extraction. The GCN model combines the features of nodes themselves and their neighbors to analyze nodes, while the disadvantage of the GCN model is that the same weight is assigned to each neighboring node, but the strength of association between different neighboring nodes is usually different. Compared with the GCN model, the GAT model allows different weights to be assigned to the same neighboring nodes, but most of the current GAT studies only have information about first-order neighbors and have not explored information about higher orders.

Graph neural networks (GNNs), which can represent an entire document and consider implicit correlations between different entities, show great potential for document-level relationship extraction. Li et al. proposed a new edge-oriented graph neural network based on document structure and external knowledge for document-level medical relationship extraction, called SKEoG, which can make full use of document structure and external knowledge [9]. In terms of GCN, Wu et al. constructed the multi-head attention graph convolutional network (multi-GCN) to improve the performance of relationship extraction [29]. Park et al. proposed an attention-based graph convolutional network (AGCN) to perform DDI (drug-drug interaction) extraction. Instead of the previous rule-based pruning, this model used a new attention-based pruning strategy [30]. A feature combination-based

graph convolutional neural (FC-GCN) model was proposed by Xu et al. [31]. Although deep models are more effective, deeper models face the vanishing gradient problem. Huang et al. [32] proposed densely connected networks (DenseNets) to solve these problems. DenseNets provide direct connections between entire layers, improving the flow of gradients and information throughout the entire layer. Additionally, they can reduce overfitting problems. The DenseNets have shown good performance in image classification tasks [33]. Zhang et al. proposed a densely connected graph attention network (IPR-DCGAT) based on iterative path inference, which uses a densely connected graph attention network to update the representation of nodes and a two-step iterative algorithm to update the representation of edges. Using the densely connected graph attention network to model local and global information between documents, the model achieves an F1 score of 84% on CDR, which is approximately 16.3%–22.5% higher than other models with insignificant margins. However, the performance degradation caused by removing the three components of distance, entity type, and co-reference embeddings is smaller than the sum of removing one of the components alone, indicating that the model may be overfitted and further model improvement is needed [34]. In addition, Guo et al. proposed a conjoined graph neural network BioGraphSAGE model with structured databases as domain knowledge, which combines bio-semantic features and location features to extract biological entity relationships from the literature [35].

### 2.1.4. Entity Relationship Extraction Based on Mixed Models

Relational extraction based on a mixed model refers to the use of different models for data processing according to the model characteristics, data characteristics, or different stages of natural language processing to achieve optimal results. A hybrid model combining RNN and CNN for biomedical relation extraction was proposed by Zhang et al. Experimental results show the complementary advantages of RNN and CNN in bio-medical relationship extraction, and the combination of RNN and CNN can effectively improve the performance of bio-medical relationship extraction [36]. Peng et al. proposed an integrated model including a support vector machine (SVM), a CNN, and an RNN, which was able to effectively detect chemical-protein relationships in the bio-medical literature and achieved the highest performance in the 2017 Challenge Task [37]. Combining the social and domain characteristics of software knowledge-community texts, entity perception information, and dependency structure information, Tang et al. proposed a model called ED-SRE, which extracts software knowledge entity relationships from unstructured user-generated content. It captures the context, semantic representation, and syntactic dependent representation of a sentence sequence using the bi-directional gating recursive unit (BiGRU) model and the GCN model, respectively. To obtain more syntactic dependence information, a weighted graph convolutional network based on Newton's cooling law was constructed by computing the syntactic dependencies between the nodes. Combining the entity type, relative entity location, and information of the entity mentioned, an entity-aware attention mechanism is proposed to integrate the entity information and syntactic-dependent information of sentence sequences to improve the prediction performance of the software knowledge entity relationship classification [10].

A sentence with overlapping relationships generates multiple conceptual instances based on different target-entity pairs. These same instances are referred to as overlapping instances. Accordingly, sentences with only one pair of target entities are referred to as normal instances. Even in the field of natural language processing, relationship extraction of sentences with overlapping relationships is a research-based difficulty. Sun et al. proposed a model with BERT representation, Gaussian probability distribution operation, and external knowledge acquisition, and experimentally found that the Gaussian probability distribution plays an important role in promoting overlapping instance extraction, outperforming location features and entity attention mechanisms in bio-medical extraction [38]. Petar Ristoski et al. proposed a method to extract positive instances of relationships from various web sources, which introduces a human-in-the-loop component in the extraction

pipeline [39]. Fossati et al. proposed the fact extractor, a complete NLP pipeline for reading the input text corpus and generating machine-readable statements, applying framework semantic linguistics theory, rather than binary techniques, to perform n-ary relation extraction [40].

### 2.1.5. Entity Relational Extraction Based on Knowledge Embedding

Most current methods of remote supervision relationship extraction focus on noise reduction processing of noisy data, while ignoring the mining and utilization of external knowledge information. Deep learning-based relationship extraction efforts are mainly dependent on word or character embeddings, regardless of the knowledge information from entities, which may lead to semantic ambiguities in the extracted relationships. Zhao et al. proposed a knowledge-enhanced relationship extraction (KERE) model using TransE and word2vec models to learn word entity information, generate knowledge-guided word embeddings, and use lexical features to enhance word semantic understanding [41]. Weinzierl experimentally found that adding lexical knowledge embeddings (LKE) or non-lexical knowledge embeddings to relationship extraction with knowledge embeddings (REKE) improved the level of relationship extraction technology [42]. A network-based attention model (NAM) for chemical-disease relationship (CDR) extraction was proposed by Zhou et al. [43]. Zhao et al. proposed a new cross-sentence n-element relationship extraction method based on self-attention, which leverages the multi-head attention and knowledge representation learned from the knowledge atlas [44]. Zhou et al. proposed an adaptive thresholding and localized context pooling (ATLOP) model for document-level relationship extraction designed to address multi-label and multi-entity problems through adaptive thresholds and local context pooling [45]. Yang et al. proposed a model called DeNERT-KG designed to extract subjects and objects via NER, and extract the relationship between them using a knowledge graph. If the proposed model has multiple entities in a sentence, each entity can be identified as a single entity and multiple relationships can be extracted; however, there is a limitation in that equivalent entities cannot be extracted into the same relationship. To overcome this limitation, techniques such as remote monitoring have been used to extract unlabeled relationships [46]. KeMRE is a medical relationship extraction method that was proposed by Qi et al. KeMRE, which predicts a relationship between TCM instructions using Chinese character sequences and medical knowledge [47]. The model is divided into four modules. First, it includes a BERT-CNN-LSTM-based text modeling framework, in which a pre-trained BERT model is used to help the model capture the semantic information of the input text better, a text CNN network for local context modeling, and a bi-LSTM network for global context modeling. A representation of each entity is obtained through careful aggregation of the role representations in each entity. Second, entities were used to model the CNN-LSTM framework to better understand the relationships between entities, using entity CNN networks to capture local associations between entities and using entity bi-LSTM networks to capture global associations between entities. Third, in the knowledge-modeling framework, to obtain medical knowledge of the drugs, the instructions for each drug were randomly selected, and the medical relationship was annotated according to the instructions. With the help of medical knowledge, knowledge embeddings were further constructed between any two entities to represent the potential relationships between them. Fourth, the relationship between any two entities is predicted based on the representations and knowledge embeddings throughout the multi-layer perceptron network (MLP).

### 2.2. Supervised Learning—Joint Extraction

Compared with the pipeline method, combining named entity recognition and relationship extraction into a single task can achieve better performance. Unlike the pipelined approach, the joint extraction framework focuses on extracting entities and relationships using a single model to capture the inherent linguistic dependencies between relationships and entity parameters, thereby solving the error propagation problem. Based on the dif-

ferent models, the joint extraction model can be divided into parameter-sharing-based, sequence-annotation-based, knowledge-enhancement-based, and graph-based methods.

### 2.2.1. Joint Extraction Based on Parameter Sharing

In the parameter sharing-based approach, by allowing two or more tasks to share the coding layer, it enables both entity recognition and relationship extraction to update the parameters of the shared coding layer during training, thus, finding the optimal parameters for the global task. The parameter sharing-based approach can effectively improve the error accumulation propagation problem and the problem of ignoring the intrinsic connection and dependency between two sub-tasks in the pipeline approach, and improve the robustness of the model [12]. At the same time, the number of parameters and the complexity of the model are reduced to achieve model light weighting while keeping the model performance unchanged. Dehghani et al. [48] showed that cross-layer parameter sharing has better performance than the standard transformer in aspects, such as language modeling. Hao et al. [49] combined the parameter-sharing transformer with the standard Hieu et al. [50] proposed efficient neural architecture search (ENAS), which achieves speedup by sharing weights for all sub-models and avoiding training from zero. Considering the long-distance relationship between entity labels, Zheng et al. proposed a hybrid neural network model without manual features to jointly extract entities and their relationships, using a bi-directional encoder-decoder LSTM (BiLSTM) module for entity extraction, and then passing entity context information to the CNN module for relationship extraction [51]. This method considers the long-range relationships between entities, but there are some relationships that are ignored and need further research to improve the recall rate. In addition, the connection between the two modules, named entity recognition and relationship extraction, needs further research to obtain better performance.

### 2.2.2. Joint Extraction Based on Sequence Annotation

The method based on parameter sharing still relies on the result of entity recognition to construct the entity pairs, and then relationship classification is used, but there is no semantic relationship between some entity pairs, which introduces redundant information to the relationship classification task. To avoid the problem of information redundancy based on parameter-sharing methods, researchers have proposed a strategy to change the triplet annotation to achieve the joint extraction of entities and relations. Zheng et al. proposed a new annotation strategy to transform the entity and relationship joint extraction task into a sequence annotation problem, where the annotated sequence information includes the position information of entity words, type of entity relationship information, and role information of entities, which can identify entities and relationships simultaneously [52]. This method uses an end-to-end neural network model to extract the relational triples between entities, which reduces the effect of invalid entities on the model and improves the recall and accuracy of relationship extraction. The advantage of this model is that it deals with isolated relationships, but the method is not effective in identifying overlapping relationships, and the association between two corresponding entities still needs to be refined. Based on joint decoding, Pang et al. proposed a deep neural network model for sequence-to-sequence-based learning called a hybrid dual-pointer network (HDP), which was designed to extract multiple-pair triples from a given sentence by generating hybrid dual-pointer sequences [53]. The performance of this method for entity overlap (one entity participating in multiple triads) is better than that for relationship overlap (multiple relationships in a pair of entities). The problem of relationship overlap is more complex, and its solution needs further exploration.

### 2.2.3. Joint Extraction Based on Knowledge Enhancement

Currently, pre-trained language models have achieved superior performance in a variety of natural language processing tasks, including entity and relationship extraction. For example, BERT achieved considerable success, but BERT-based models do not perform

as well on Chinese domain-specific corpora as on English data sets. When BERT is used with a Chinese corpus, it generates embeddings at the character level; however, Chinese vocabulary contains more semantic information than Chinese characters. Therefore, BERT-based models are limited in their ability to extract dependencies between Chinese words, which are very important in relationship extraction. Ding et al. proposed an extraction architecture for joint entities and relationships to address this problem using external lexical and syntactic knowledge to overcome the limitations encountered by BERT-based models in the joint extraction process of specific Chinese fields [54]. The framework implements a knowledge-enriched and span-based BERT (KSBERT) based on knowledge-rich and span networks, which combines the dependency structure to extract entities and their relationships simultaneously. An end-to-end model of knowledge-enhanced word-embedding BERT (KEWE-BERT) for joint entity and relationship extraction was proposed by Dong et al. [55]. This model can improve the effect of knowledge extraction in the case of limited domain corpora and has good portability. Yadav et al. developed an end-to-end knowledge injection deep learning framework (Gated-K-BERT) that utilized a pre-trained BERT language representation model and domain-specific declarative knowledge sources (drug abuse ontology), using a gating, fusion, and sharing mechanism to jointly extract entities and their relationships. Combining knowledge-aware attentional representations with BERT extracts broader cannabis-depression relationships [56]. The model combines entity knowledge in the form of entity location-aware encoding with attention, which helps to perform better relationship classification. Experiments show that location-aware encoding and location attention have a significant impact on the effectiveness of relationship extraction, and the model performance decreases if they are missing. The limitation of the model is that it experiments with tweets on Twitter with short text, which may not be applicable to large text.

### 2.2.4. Joint Extraction Based on Graph Structure

Compared with the pipeline method, the above joint extraction method avoids the problems of error accumulation and poorly connected sub-tasks in the traditional method, takes into account the dependency between two tasks, and improves the overall task accuracy. However, it cannot deal with the overlap problem in relationship extraction well. More and more researchers are exploring the use of graph neural networks to solve the overlap problem, of which GCN and GAT are typical.

Fu et al. proposed an end-to-end relationship extraction model based on graph convolutional networks (GCNs) that jointly learn named entities and relationships. The role between named entity recognition and relationship extraction is considered by a relation-weighted GCN. The model combines RNN and GCN to extract not only sequence features of each word but also region-dependent features using linear and dependency structures, and the relationship of each word pair is predicted using the complete word graph, considering implicit features between all word pairs in the text, solving the problem of entity overlap [57]. Miao et al. addressed the problems of error propagation in relationship extraction, redundancy in prediction, and inability to solve the relationship overlap problem. They proposed a joint entity relationship extraction model BSGB (BiLSTM + SDA-GAT + BiGCN) based on graph neural networks, which extracts the sequence features of each word and the local dependency features of each word, and employs a graph convolutional network that considers the implicit features between all word pairs in the text to predict the relationship of each word pair, thus, solving the overlap problem [58]. Qiao et al. proposed a joint entity-relationship extraction model based on graph convolution-enhanced multi-channel decoding (graph convolution-enhanced multi-channel decoding joint entity and relation extraction, GMCD-JERE); the next step can consider how to improve the extraction performance of the model in the case of insufficient samples and uneven distribution and explore solutions for long or cross-sentence entity relationship extraction [59].

### 2.3. Entity Relational Extraction Based on Distant Supervision

Existing neural-network-based methods have achieved great success. However, most supervised relational extraction models require large amounts of training data, which are expensive to obtain. To overcome this shortcoming, remote supervision has been introduced to construct large-scale data sets automatically.

KIM et al. proposed a remote supervision-based model that requires no annotation and can be used to represent the context of a sentence and the relationships mentioned around it, thus, enabling paragraph-level relationship classification [60]. Zeng et al. proposed a new segmented convolutional neural network (PCNNs) model using multi-instance learning for remote supervised relationship extraction [61]. Deng et al. proposed a note of knowledge representation designed to utilize the knowledge information in a knowledge database to reduce the impact of noisy data on remote supervised relationship extraction. The distributed representation of the knowledge database is pre-trained using the knowledge representation learning (KRL) model, and then included in the relationship extraction to learn the sentence-level attention weights. Attention is focused on valid data by utilizing background information in the knowledge base [62]. In this approach, the quality of knowledge embedding is crucial in this model. Most LSTM-based models only learn word representations and cannot represent semantic blocks. Many studies have focused on single-relation extraction and have ignored multi-relationship extraction. Huang et al. divided sentences into solid and nonsolid blocks and proposed block graph LSTM networks to simultaneously learn the representations of entities and relationships. The block LSTM, along with the graph LSTM, was integrated into the LSTM network for multi-relationship extraction [63]. A network relation extraction method based on remote supervised classes strategically selects seeds for training, extracts relation mentions across sentence boundaries, and integrates relation mentions to predict relations of knowledge-based groups [64]. Lin et al. proposed a remote relationship extraction model trained to identify distant relationships using bootstrapped noise data, combining the article chapter structure, graphical information, and attention mechanisms proposed by Lin et al. [65]. Huang et al. proposed a novel GCN-based local-to-global graph convolutional network model called LGGCN, which encodes sentences in packets from a local to global perspective to improve the performance of remote-supervised relationship extraction [66]. A new remote supervised relationship extraction (DSRE) framework based on an adaptive dependent path and additional knowledge graph supervision called $A^2$DSRE was proposed by Shi et al. An advanced graph neural network, GeniePath, was introduced as an adaptive path layer in DSRE, and TransE was used to obtain the relation embedding from the knowledge graph as additional supervision [67].

Mao et al. proposed a new framework for a remote-supervised relationship extraction task using a knowledge-attention-guided graph convolutional network, which consisted of two modules: a sentence embedding module and a multi-instance selection module [68]. The sentence embedding module uses the relational indicators in the lexical resources as prior knowledge to guide sentence embedding, which consists of a word-level knowledge attention layer and a graph convolutional layer. It uses relationship indicators obtained from the framework network to effectively capture information-rich language clues and generate more expressive sentence features. The multi-instance selection module uses the structure and semantic information in the knowledge graph as prior knowledge to guide the selection of multiple effective sentences, including knowledge graph embeddings and sentence-level knowledge attention layers. The model utilizes hierarchical knowledge attention to focus on multiple instances to mitigate different degrees of noise, thus, generating more expressive relationship representations to enhance the relationship extraction. Distant supervision has been demonstrated to be highly beneficial for enhancing relationship extraction models, but it often suffers from high label noise. Wang, Z., et al. propose a novel model-agnostic instance sub-sampling method for distantly supervised relationship extraction, namely REIF, which bridges the gap between realizing influence in sub-sampling in deep learning [69]. It encompasses two key steps: first, calculating

instance-level influences that measure how much each training instance contributes to the validation loss change of the model, then deriving sampling probabilities via the proposed sigmoid sampling function to perform batch-in-bag sampling.

*2.4. Summary*

As mentioned in this section, the pipeline extraction-based method is relatively straightforward to implement; the flexibility of the extraction models is high; and the entity and relationship models can use independent data sets without requiring both annotated entities and relations. However, it involves the following shortcomings: 1. the error accumulation affects the performance of the next relationship extraction, causing error propagation, 2. because the extracted entities are paired and then classified, the redundant information of un-related candidate entities increases the error rate and increases the computational complexity, and 3. the missing interaction ignores the internal connection and dependence between the two tasks. The joint extraction framework focuses on extracting entities and relationships using individual models to capture the linguistic dependencies inherent between relationships and entity parameters, thereby addressing some problems of pipeline extraction. Some progress has been made in existing methods of remote-supervised relationship extraction, whose paradigm can automatically collect training data for relationship extractors but often encounters incorrect labeling problems. Remote monitoring inevitably introduces noise to the resulting training data set, and thus reduces the relationship extraction performance. They still face two challenges, including designing more efficient sentence encoders to generate more expressive sentence-level features, and figuring out how to make full use of the informative sentences in the package and then integrating them to generate package-level features to predict a given relationship.

## 3. Event Extraction Based on Deep Learning

Anything occurring in a real event can be considered as an event, and people understand the world by understanding the relationship between events. An event occurs at a specific time and in a particular place, involving one or more participants, and can often be described as a change in state. Event extraction is a sub-task of IE that plays a very important role in the field of knowledge mining. It aims to extract events of interest from unstructured information, identify specific types of events, and present the elements of the established role in a structured form. Event extraction can be further decomposed into four sub-tasks: trigger word recognition, event type classification, meta-identification, and role-classification tasks.

Event extraction is a deep research topic in the field of IE, which is based on research on entity recognition and relationship extraction. In recent years, to construct event extraction system-related research developments, the traditional event extraction method, namely, based on the mode-matching method, required field qualification to be excessively strong, the effect greatly depended on the data dimension quality and annotation scale factors, artificially built templates were difficult to create, the connection between multiple events was difficult to model, and event authenticity detection was not performed. Building an effective event extraction system from scratch suffers from several difficulties and obstacles. In contrast to traditional methods based on pattern matching, deep learning methods based on neural network models have attracted increasing attention from researchers. Its general logic is to represent the text sequence as a computable multi-dimensional tensor and to realize the classification of event-triggered words and event elements by constructing end-to-end deep learning models. A deep learning method based on the neural network model uses word embedding to represent text, which does not require manual specification of the features and greatly improves the effect of the event extraction task [70]. Embedding represents hard matching of feature engineering with soft matching, and features are automatically extracted using linear and non-linear functions in the network. Neural network models have strong combinatoriality. Transfer learning can combine different models reasonably to learn from each other, which mitigates problems, such as error

propagation and implicit feature loss in feature learning. The model based on multi-task learning can predict event trigger words and event elements simultaneously, realize parameter sharing between tasks, use the correlation of the two tasks to improve each other, and finally improve the event extraction effect overall. This chapter summarizes event extraction from two aspects according to the different learning modes and scopes of the model.

### 3.1. Single-Task or Multitask Event Extraction

Neural network event extraction is divided into single-task extraction, according to different model learning modes, and multi-task extraction. Usually, event extraction can be summarized into tasks, such as trigger word extraction and argument extraction. The single-task extraction model transforms the event extraction task into a multi-stage classification problem, and the extraction is completed sequentially or jointly. Chen et al. proposed a dynamic multi-pooling convolutional neural network (DMCNN), as shown in Figure 5, which was designed to divide a sentence into multiple modules through sequence division and successively perform the tasks of trigger word extraction and argument extraction [71]. In the event extraction task, because a sentence may contain multiple events, different arguments play different roles in different event types, and the traditional convolutional neural network model using the maximum pooling in each sentence can capture the maximum multiple-event sentence event extraction models misinformation. The DMCNN uses dynamic multi-pooling convolution to capture different parts of a sentence, thereby retaining more critical information. In an event, the amount of argument information is far greater than that of the trigger word, and because the pipeline model is used in the extraction, the LIU joint model primarily solves the problem of argument recognition extraction. Through a series of experiments, it was found that the joint model for the detection of the argument prediction effect was significantly better than the effect of argument detection alone [72]. Since the meta-information in the sentence is more important than the non-meta-information, the study adopted the attention mechanism to improve the weight of the meta-information when triggering word detection, and the results achieved a good detection effect. As a representative work, Nguyen et al. used a bi-directional recursive neural network (JRNN) for event extraction (as shown in Figure 6). This method reduces the propagation of the error by combining local and global features in the recurrent neural network and designing different memory matrices to enhance the connection of different events [73]. Nguyen et al. compared their JRNN model with the CNN and DMCNN models mentioned above and found that this model significantly outperformed other models [71]. As shown in the following figure, the model is more accurate than the DMCNN model in trigger word recognition, further indicating that the JRNN benefits from a memory function. For the event extraction task, trigger word recognition is the basis of the task, and its results will have an impact on subsequent work; thus, the JRNN model achieves good results.

In the single task of joint event extraction, considering the correlation of named entity identification and event detection, an increasing number of researchers have proposed joint learning-based event detection methods to extract events and entities simultaneously. Multi-task learning refers to a given machine learning model endowed with multiple sub-tasks. Each sub-task has a certain correlation and improves the learning ability of the model on multiple tasks by using the correlation between each sub-task. Compared with single-task models, multi-task models have shown better results in named entity recognition tasks and event detection tasks. From the perspective of the multi-task sharing mode, the existing multi-task learning model can be divided into three methods, including a hard sharing, soft sharing, and sharing-private modes [74]. The hard sharing mode uses the same representation shared between multiple tasks, with the private layer adapted to a specific task. The soft sharing mode indicates that after the sharing layer, the sharing layer output produces different ratios. For the same input, each task obtains different representations with a certain flexibility. The sharing-private mode uses a private representation of each

task in addition to a shared representation. When entering the private network of each task, the shared representation and the private representation are spliced for each task.

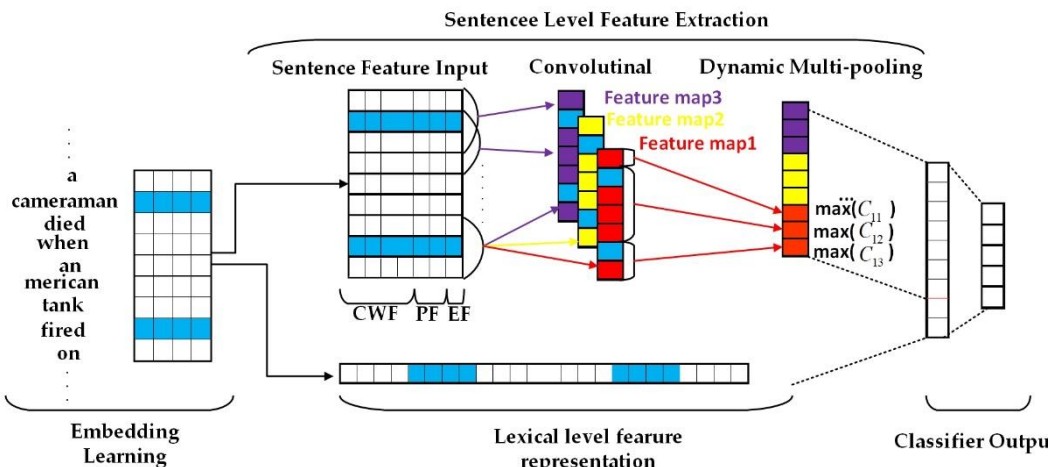

**Figure 5.** The DMCNN uses dynamic multi-pooling convolution to capture different parts of a sentence, thereby retaining more critical information [71].

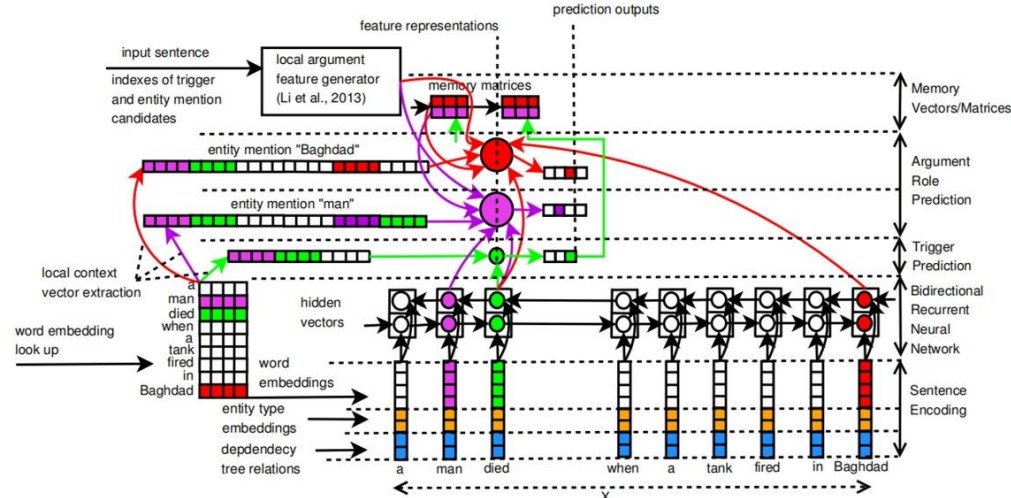

**Figure 6.** The architecture of JRNN, which reduces the propagation of error by combining local and global features in the recurrent neural network and designing different memory matrices to enhance the connection of different events [73].

From the perspective of multi-task applications, the application of multi-task learning to domain migration scenarios has received considerable attention in recent years. Kru-engkrai et al. proposed a multi-task model, with named entity recognition as the primary task and sentence classification as an auxiliary task, and improved the performance of the low-resource NER model by jointly training the NER and sentence classification models [75]. Most existing entity link tasks focus only on entity disambiguation while ignoring entity recognition. Martins and others jointly learned named entity recognition and entity links and added an attention mechanism to Stack-LSTM. Each decision uses the information from the two tasks to obtain a more robust system [76]. He Ruifang et al. first annotated triggers and event elements jointly. To avoid multiple triggers for a sentence, the model trained an extraction model for each event type. Finally, to solve the problem of data sparsity, multi-task learning sharing task information was proposed [77]. To solve the problem of low-resource language sequence annotation, as an example, Lin et al. proposed a multi-language and multi-task framework, the infrastructure of which comprises CNN encoding character information and an LSTM encoding context information using the conditional

random field (CRF) model to obtain the optimal sequence labels. This method shared the coding and linear layers and the CRF layer [78]. Yu et al. used the correlation between the two and proposed the MDL-J3E joint extraction model based on multi-task learning, which is essentially a shared-private primary-auxiliary multi-task learning network model that regards event detection as the main task and named entity recognition as an auxiliary task [74]. An empirical study on the ACE2005 data set showed that the proposed model achieved 84.15% on the named entity recognition task for F1 and 70.96% for the event detection task (The results are shown in Table 3). Compared with single-task learning, multi-task learning can fully alleviate the problem of sparse data, solve the problem of the existing event extraction model by assuming that entities exist or ignoring entity information, and provide a new idea for the event extraction task. Wang et al. proposed a novel named entity recognition model framework containing shared-private domain parameters and multi-task learning applied to multi-domain and domain labels not encountered in training [79]. The model adopted the basic framework of BiLSTM-CRF and modelled the feature mapping of shared and private features based on this framework. Private modules were used when domain labels and common modules' domain labels were unknown. In addition, domain recognition was also included as an auxiliary task for named entity recognition tasks to help the model further improve its performance. In addition to domain-transfer tasks, multi-task learning is widely used in areas, such as law, medical care, and emotion analysis. In addition, Yaojie Lu and others creatively proposed a unified text-to-structure generation framework named universal information extraction (UIE), which can model different IE tasks in general, generate target structures adaptively, and learn common IE capabilities from different knowledge sources (as shown in Figure 7). The experimental results show that UIE has achieved very competitive performance in both supervised and low-resource environments, which verifies its versatility, effectiveness, and portability [80].

**Table 3.** The effects of DL models.

| Model | NER | | | Event Extraction | | |
|---|---|---|---|---|---|---|
| | P/% | R/% | F1/% | PI% | R/% | F1/% |
| Layered-BiLSTM-CRF | 74.20 | 70.30 | 72.20 | - | - | - |
| GEANN | 77.10 | 73.30 | 75.20 | - | - | - |
| BiFlaG | 75.00 | 75.20 | 75.10 | - | - | - |
| Merge and Label [ELMO] | 79.70 | 78.00 | 78.90 | - | - | - |
| Merge and Label [BERT] | 82.70 | 82.10 | 82.40 | - | - | - |
| JOINTEVENTENTITY | - | - | - | 75.10 | 63.30 | 68.70 |
| DMCNN | - | - | - | 75.60 | 63.60 | 69.10 |
| FN-ANN | - | - | - | 79.50 | 60.70 | 68.80 |
| BDLSTM-TNNs | - | - | - | 75.30 | 63.40 | 68.90 |
| JRNN | - | - | - | 66.00 | 73.00 | 69.30 |
| TD-DMN | - | - | - | 65.80 | 65.90 | 65.60 |
| RNN-AL | - | - | - | 77.40 | 61.30 | 67.80 |
| GAIL | - | - | - | 74.20 | 65.30 | 69.50 |
| Conv-BiLSTM | - | - | - | 74.70 | 64.90 | 69.50 |
| ANN-Gold2 | - | - | - | 81.40 | 66.90 | 73.40 |
| HNN-EE | 84.00 | 82.50 | 83.20 | 74.40 | 67.30 | 70.60 |
| Single task NER (MDL-J3E) | 83.86 | 84.10 | 83.98 | - | - | - |
| Single task ED (MDL-J3E) | - | - | - | 66.67 | 74.25 | 70.25 |
| MDL-J3E | 83.48 | 84.83 | 84.15 | 69.16 | 72.85 | 70.96 |

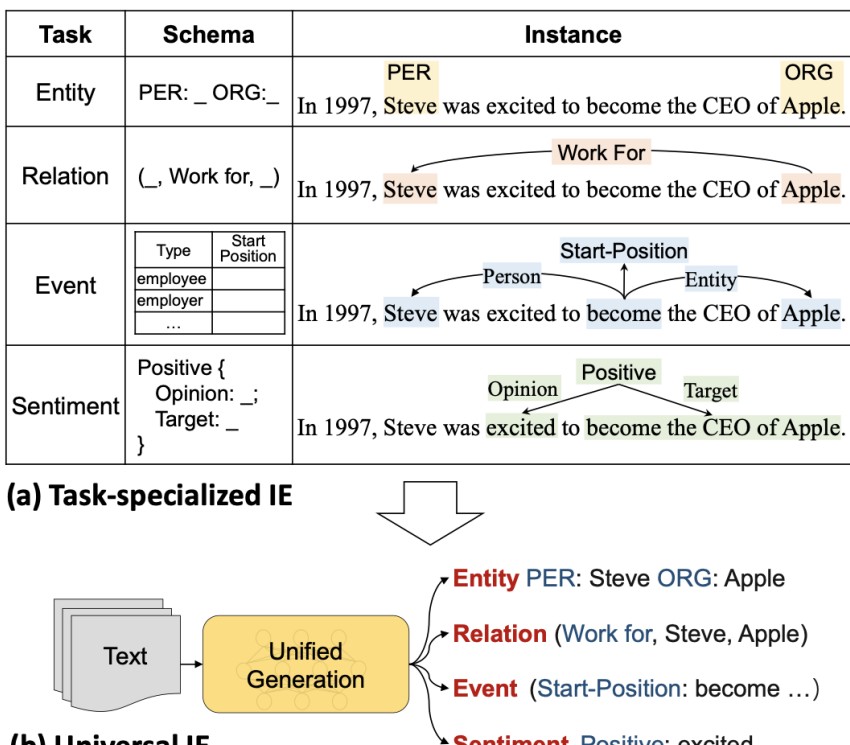

**Figure 7.** From (**a**) task-specialized IE: different tasks, different structures, different schemas to (**b**) universal IE: unified modeling via structure generation [80].

In specific fields, researchers have mostly used the joint extraction method to perform event extraction. For example, in the field of TCM, Gao et al. used the joint event extraction model to divide the model into four layers: (1) a BERT layer representing text, (2) a BiLSTM layer learning an input vector, (3) a self-attention layer mainly used to capture the dependence between words, and (4) a CRF layer that output the results [81]. Li et al. proposed a joint financial event extraction method integrating the pre-trained model and multi-layer convolutional neural network (BERT-MultiCNN), which captures the semantic information in events and further improves the effect of event extraction in this field [70]. Yu et al. extracted the event information contained in ancient texts, selected Zuo Zhuan as the data set, and used the RoBERT-CRF model. However, the subject category could not be balanced owing to a small data size [82]. Event extraction requires high-quality expert human annotations, which are usually expensive. Hsu, I. H. et al. focus on low-resource end-to-end event extraction and propose DEGREE, a data-efficient model that formulates event extraction as a conditional generation problem. Given a passage and a manually designed prompt, DEGREE learns to summarize the events mentioned in the passage into a natural sentence that follows a pre-defined pattern [83].

### 3.2. Sentence-Level or Chapter-Level Event Extraction

Generally, for simple events, types of events can be directly identified from a sentence and the required metadata can be extracted. However, with the escalating complexity of the event, sentence-level extraction cannot cover all the metadata of the event, and complete information needs to be extracted from multiple sentences. Therefore, the event extraction task can be divided into sentence-level and chapter-level extractions from the learning scope. The research task of event extraction was initially based mainly on sentence-level extraction of ACE2005 data. Gradually, the information obtained from the extraction of the isolated sentences was likely incomplete.

For sentence-level extraction, Feng et al. combined a mixed neural network with a BiLSTM model and a CNN model to conduct event extraction by modeling the sequence of

input sentences and obtaining semantic information of the structure and sequence from a specific context [84]. Miao et al. introduced a new event-triggered word extraction model, CNN-BiGRU, which extracted word-level features through CNN, then captured text semantic information through BiGRU to obtain sentence-level features, and finally spliced word-level and sentence-level features to achieve the identification of trigger words and predict event categories [85]. Wu et al. proposed a joint method for extracting entities and events in sentences using the HNN-EE model [86]. The model uses the BiLSTM model to identify entities, and then transmits the context information acquired on the BiLSTM to the self-attention layer and gating convolutional layer of the neural network for event extraction, as shown in Figure 8.

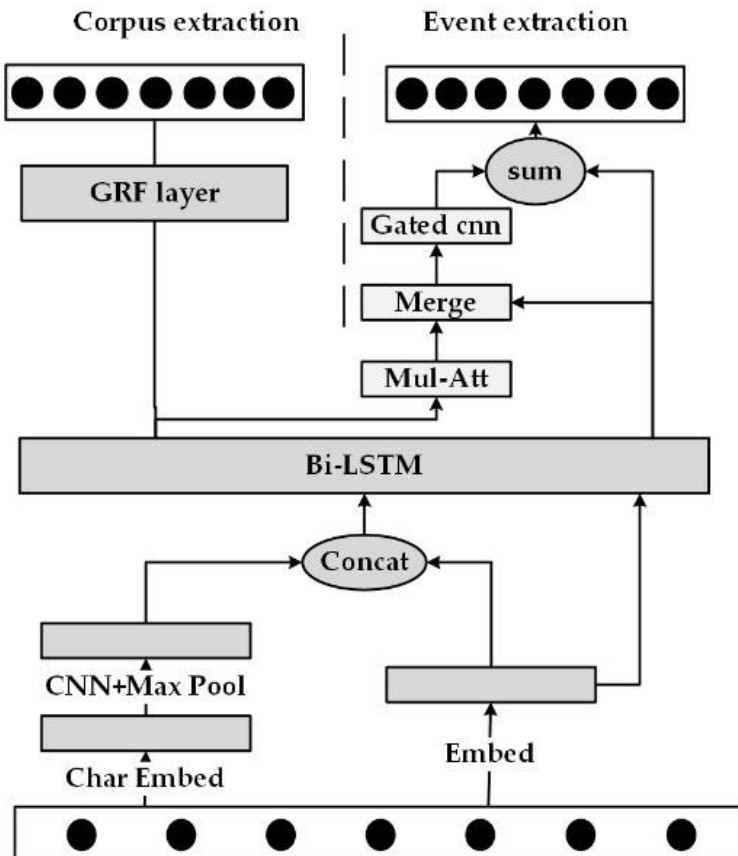

**Figure 8.** The architecture of HNN-EE, which uses the BiLSTM model to identify entities, and then transmits the context information acquired on the BiLSTM to the self-attention layer and gating convolutional layer of the neural network for event extraction [86].

For document-level event extraction, the entire document is directed as the input, and then the desired event elements are the outputs. Chapter-level event extraction requires new ideas. In a document that describes an event, an event-centric sentence often expresses the event best. Most of the information is often covered in the event center sentence; therefore, Yang et al. constructed a DCFEE framework for document event extraction according to the above idea. The specific architecture of the framework is shown in Figure 9. The framework was designed to divide the event-extraction process into two parts. First, using sentence-level extraction, the BiLSTM-CRF sequence annotation model was used for the input sentences to obtain the trigger words and arguments of the sentence. The second part performed document-level extraction. If the sentence was judged as being central to the output of the previous step, the arguments were extracted in the context of a sentence with full event information [87]. Zhong et al. elucidated the differences between sentence-level event extraction and document-level event extraction, and jointly extracted sentence-level entities based on the attention mechanism sequence annotation model. Then,

based on sentence-level event extraction, sentence-level information was integrated using integer linear programming to obtain document-level event information [88]. Zheng et al. proposed a novel end-to-end event extraction model for the financial domain [89]. The main idea of the model was to convert document-level event table filling (DEE) into an entity-based directed acyclic graph (EDAG), which can be divided into four steps: entity recognition, document-level information coding, EDAG generation, and EDAG path expansion. Moreover, the end-to-end learning model was able to extract more comprehensive features and reduce error propagation through joint learning. Guo et al. constructed a COVID-19 news data set and proposed a three-stage pipeline approach to extract COVID-19 news events from the chapters [90]. The improved TextRank algorithm extracted the central sentence in the document and then used sequence annotation to extract the events from the chapter-level perspective to obtain more complete event information. Yang et al. proposed a joint framework that trained the three sub-models to learn the event internal structure, the relationship between events, and entity extraction, and then integrated them into a single model to realize joint extraction of events and entities across documents [91].

In addition, some researchers have put forward a new idea of event extraction. Liu, Chen et al. translated the event extraction task into a Q and A. The author's idea was to identify the event type through the task of event extraction, generate the question template of the event elements in an unsupervised manner according to the type of event, and finally extract the event elements in the form of questions and answers [92]. Inspired by the English event extraction method using the question answer mode, Liu et al. applied the method to the task of Chinese event extraction and designed a set of generation rules that meet the problem template of Chinese event extraction [93].

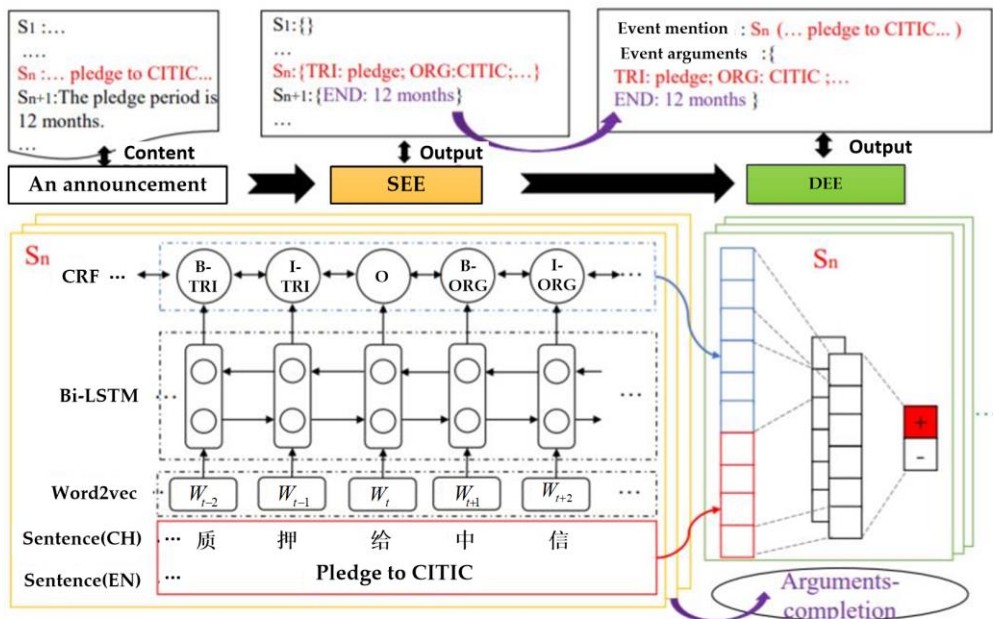

**Figure 9.** The Architecture of DCFEE, which was designed to divide the event-extraction process into sentence-level extraction and document-level extraction into two parts [87].

### 3.3. Summary

As mentioned in this section, in terms of the accuracy and prediction effect of trigger word recognition and argument recognition, researchers currently prefer joint multi-task extraction methods. The experimental results have also proven many times that, compared with the pipeline or single-task extraction method, the multi-task joint extraction method causes trigger words and argument information exhibits a mutually reinforcing extraction effect. Since the information obtained from the extraction of isolated sentences is likely to be incomplete, researchers have shifted their focus to chapter-level extraction. However,

the accuracy of many experimental results does not meet the requirements of practical applications. However, there are still many other issues that need to be addressed. For example, in the current study of event extraction, most were extracted based on the existing annotated corpus. There are few studies and poor results, and current methods are only useful for specific event types. Therefore, establishing more standard and mature corpora and improving the extraction effect of unlabeled corpora are worthy of further research. For chapter-level event extraction, how to effectively integrate the information extracted from different sentences is also a major challenge to be solved, which requires researchers to overcome the lack of semantic understanding techniques within and across chapters. In addition, system performance and portability are the two most important factors restricting the widespread application of event extraction technology. Future research can focus on how to overcome and solve these two problems, continuously improve the performance of the event extraction system, and enhance its portability.

## 4. Multi-Model IE Based on Deep Learning

Multi-modal information extraction is the combination of multi-modal learning and information extraction technology [94]. Traditionally, the study of IE has focused on extracting entities and relationships from pure text, where information is mainly represented in the format of natural language text [95]. However, the rapid development of the Internet has resulted in massive amounts of data, including text, audio, images, video, and other modalities. Multi-modal information on the Internet, in some scenarios, only for the text of the data information extraction, may cause the loss of data information; therefore, researchers began to discuss how to extract the required information from multi-modal data. Existing work has demonstrated that the addition of visual modal information can play an important role in work, such as knowledge graph completion and triplet classification, and that multi-source information has shown potential for reasoning on the knowledge graph.

"Modality" is widely defined, which can be intuitively understood as different types of multi-media data or as a more fine-grained concept. The key point to distinguishing modalities can be understood as whether data are heterogeneous. For example, for an actor, relevant information can be found on the Internet, including text introductions, personal pictures, film and television works, and film and television audio. These four types of data correspond to text, picture, video, and sound, respectively, which can be understood as multi-modal data of the object [96]. In a multi-modal data environment, cross-modal data have both modal characteristics and semantic commonalities. Multi-modal IE is a combination of multi-modal learning and IE technologies. At present, the existing unimodal representation learning methods have achieved good results, laying a foundation for the acquisition of multi-modal representations. The development of deep learning has also provided convenience for multi-modal research. The following two subsections consider multi-modal named entity recognition and multi-modal relationship extraction.

### 4.1. Multi-Modal Entity Identification

Traditional named entity recognition only considers text information and ignores the influence of integrating other modes on the recognition of named entities. In view of the deficiency of using unimodal information to identify named entities, scholars have begun to study the task of named entity recognition combined with multi-modal information. Most multi-modal methods use the attention mechanism to extract visual information but ignore whether there a correlation exists between the text and the images, and text-independent visual information can have uncertain and even negative effects on the learning of multi-modal models. Sun et al. proposed a multi-modal BERT model based on a text-image relationship propagation-based multimodal BERT model (RP-BERT) for text-image relationship classification (TRC), and trained the model RP-BERT on the MNER. The model achieved the highest F1 scores for both the TRC and MNER. Experimental results showed that the propagation of text-image relationships was able to reduce the interference of irrelevant images, and RP-BERT makes better use of visual information based on the text-

image relationship [97]. Zhang et al. constructed and obtained a large-scale labeled dataset containing multi-modal tweets from Twitter. To use visual information to identify named entities in multi-modal tweets, Zhang proposed an adaptive co-attention network (ACN) linking text and visual information (as shown in Figure 10) [98]. An adaptive co-attention network layer was inserted between the hidden and CRF layers to pay mutual attention to the representation of text and pictures; thus, each word acquired a multi-modal representation by introducing a gated multi-modal fusion module to decide when to rely on visual information. A filter gate module was also used to filter the noise caused by visual information. The model introduced image information based on CNN + BiLSTM + CRF, adding an ACN module on the constructed dataset with 72.75% accuracy, recall of 68.74%, and 70.79% F1, with better performance than CNN + BiLSTM + CRF. Some MNER models do not take full advantage of the fine-grained semantic correspondence between different modal semantic units, which may optimize multi-modal representation learning. A unified multi-modal graph-fusion (UMGF) approach for MNER was proposed by Zhang et al. [99]. A unified multi-modal graph was first used to represent the input sentences and images. During the composition process, each target image acts as an image node. Each word acted as a text node. The graph captured the various semantic relationships between modal semantic units (words and visual objects). After stacking multiple graph-based multi-modal fusion layers, semantic interactions were iteratively performed to learn the node representation. Using a graph neural network to interact with the two modal units, a further two-stream version of the cross-modal gating was used. Finally, passing through a linear layer and CRF encoding layer obtained the final output. In the experiments on two benchmark datasets, the F1 values of this model were higher than the F1 values of the other methods, and UMGF performed better for multi-modal named entity recognition.

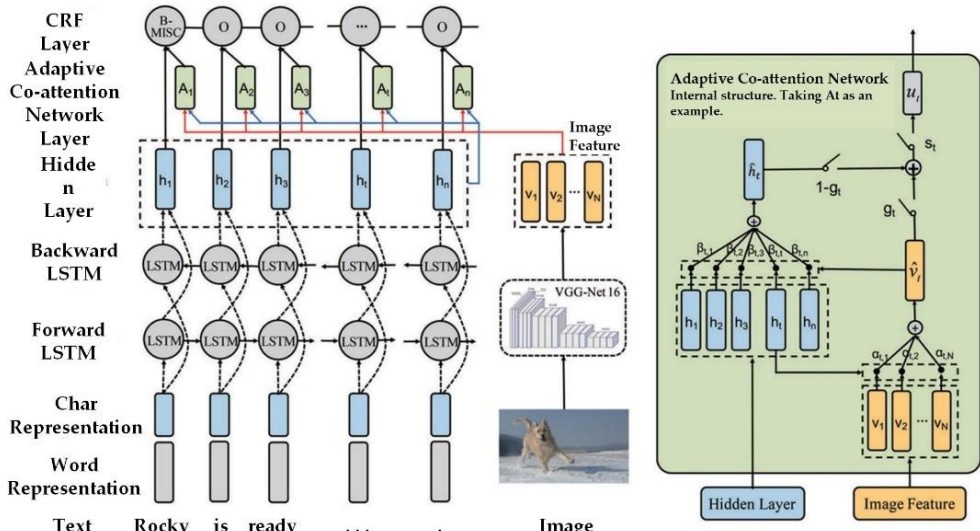

**Figure 10.** The general architecture of ACN. An adaptive co-attention network layer was inserted between the hidden and CRF layers to pay mutual attention to the representation of text and pictures [98].

At present, great progress has been made in multi-modal named entity identification, but most of the studies have focused on English, and most of the previous studies on Chinese named entity recognition have focused on unimodal text. Sui et al. studied Chinese multimodal named entity recognition from both text and acoustics, constructing large-scale manually annotated multimodal named entity recognition data (CNERTA) with text and acoustic content [100]. Based on this dataset, a series of baseline models were established, including BiLSTM-CRF and BERT-CRF, which can use text-modal or multi-modal features. In addition, by introducing a speech-text alignment auxiliary task, a simple multimodal multi-task model (M3T) was proposed to capture the natural monotonic

alignment between text and acoustic modes. As shown in Figure 11, in the M3T model, acoustic information is integrated into the text representation using a cross-modal attention module (CMA). Through extensive experiments, the authors demonstrated that Chinese named entity recognition models can benefit from introducing acoustic modes.

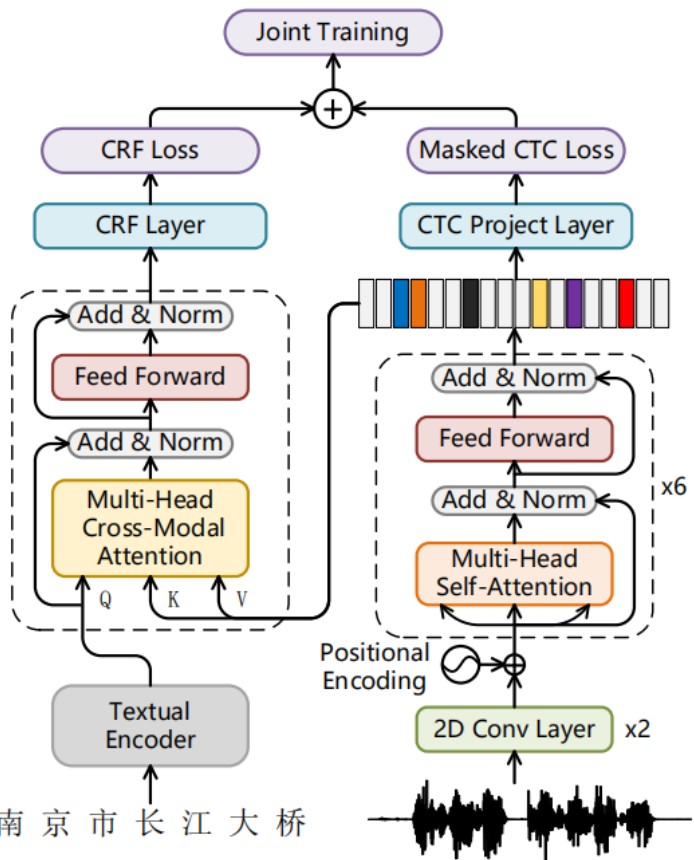

**Figure 11.** M3T was proposed to capture the natural monotonic alignment between text and acoustic modes [100].

### 4.2. Mult-Imodal Relationship Extraction

In many cases, adding corresponding entity pictures can improve the recognition effects of the relationship between entities. When providing two corresponding pictures, the model can be based on the corresponding relationship between the two; otherwise, if there is no similar sample training set, it will be difficult to judge the relationship between the two. Xie et al. proposed a new image-embodied knowledge representation learning model (IKRL) designed to integrate multi-modal information to improve the accuracy of triplet prediction [101]. The IKRL model is shown in Figure 12. Xie et al. first proposed an image encoder consisting of a neural representation module and projection module, taking the images formed by each entity as a feature input. Second, attention-based instance-level learning methods were used to automatically calculate their attention to different image instances. Finally, image-based aggregate representations were jointly learned with structure-based representations using the overall energy function. IKRL models for evaluating knowledge graph completion and triplet classification using WN9-IMG. In terms of entity prediction, such as the translation-based approach of TransE, which only considers structured information in the triplet, it ignores other information and may fail to predict relationships. However, the image information used in IKRL can provide supplementary information for better entity prediction. In terms of triplet classification, the model combines ternary structure information and visual information in the image and achieved an accuracy rate of 96.9%, which was higher than 95.0% for Trans E and 95.3% for

TransR. The experimental results demonstrated the importance of visual information for knowledge representation and the ability of the model to learn knowledge representation using images. Sergieh et al. further improved Xie et al. in that model, in addition to images, integrates an external representation of the language embedding of knowledge graph entities, enriching the representation of entity text modalities [102]. After merging the corresponding text representation of the entity with the image representation, a multi-modal representation of the entity was obtained. Based on multi-modal translation methods, the energy of KG triples is defined as the sum of the sub-energy functions using multi-modal (visual and language) and structural KG representations. In triplet classification, the proposed model achieves better results on the WN9-IMG and FB-IGM datasets.

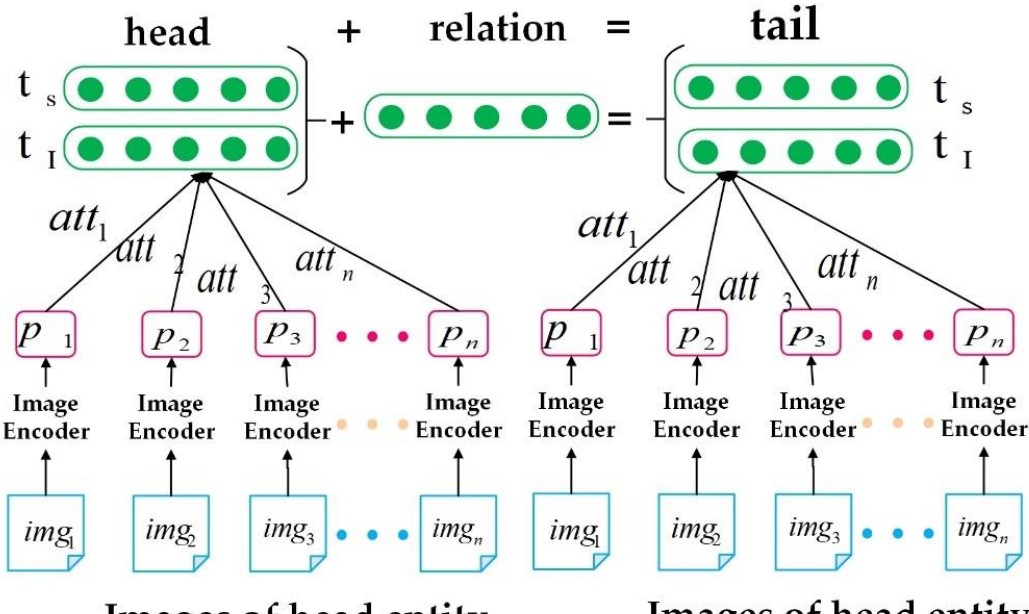

**Figure 12.** Overall architecture of the IKRL model, which integrates multi-modal information to improve the accuracy of triplet prediction [102].

In social media posts, traditional relationship extraction methods exhibit significantly lower performance when the text is short and lacks context; however, the images associated with these sentences can complement the missing context and help accurately identify relationships. To solve this problem, a multi-modal neural relation extraction (MNRE) dataset was proposed by Zheng et al. [103]. First, the entities and their corresponding types were extracted using the trained NER labeling tool, and then the relationships between the entity pairs were manually labeled. A labeling tool was developed that could simultaneously display each entity pair in a sentence and related images. The love + CNN, BERT NER, BERT + CNN, and PCNN models were selected for the experiments, and image labels, visual objects, and visual attention were added as the baselines for multi-modal relationship extraction. The experimental results showed that the overall performance was improved when bilinear attention was added to obtain the correlation between text and vision (except for PCNN), and introducing multi-modal information into social media text can improve the performance of relationship extraction. Subsequently, Zheng et al. developed a high map alignment to learn from a multi-modal neural network with efficient graph alignment (MEGA) to learn the visual relationship corresponding to the text relationship [104]. The Overall Framework of MEGA Model is shown in Figure 13. The dual graph alignment method can capture the correlation between visual and text, combine the structural similarity and semantic consistency between visual objects in images and text entities in sentences, and find the most similar nodes between the two graphs with structure and semantic features to better align the text and visual relations and use visual

relationships. Experimental results on the M NRE dataset show that the introduction of visual information can complement the missing semantics of social media texts, and efficient graph alignment methods can find correlations between visual and language, resulting in better performance.

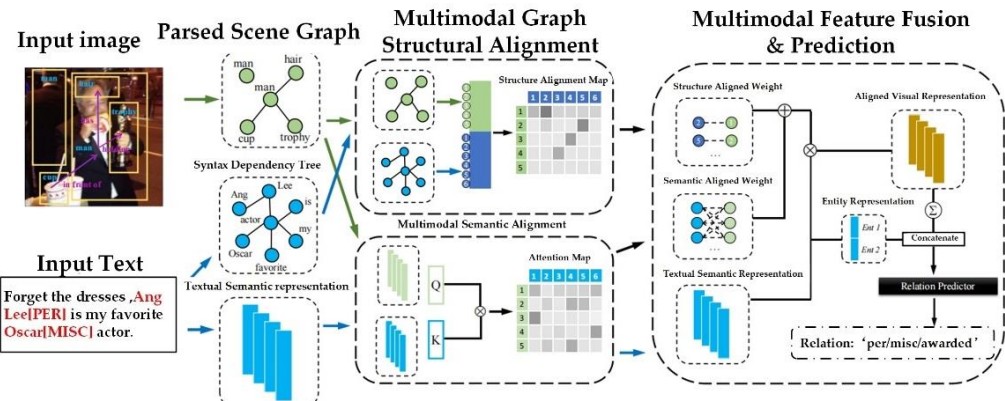

**Figure 13.** The Overall Framework of the MEGA to learn the correspondence between vision and text [104].

The ability to jointly express the semantics of information and document type is a new area of IE research in which visual and textual information play an important role in its analysis and understanding. Visually rich documents (VRD) are ubiquitous in daily business and life. The semantic structure of visually rich documents depends not only on the text, but also on their layout, table structure, and font size. There is still much information untapped in visually rich documents; the visual and layout information is critical for document understanding, and the text in such documents cannot be serialized into one-dimensional sequences without losing information. In addition to the rich visual effect and the nature of text, Oral et al. proposed a new graph decomposition-based relationship extraction algorithm to solve the relationship in the document for the n yuan, nested, document level, and previously uncertain number of complex relationship extraction problems using optical character recognition technology to extract image text components and cross-media natural language processing technology for IE [105]. A graph-convolution-based model combining textual and visual information in VRD was presented by Liu et al. [95]. The graph convolution-generated graph embeddings summarize the context of the text segments in the document and are further combined with text embeddings using a standard BiLSTM-CRF model for entity extraction.

However, existing approaches for MNER and MRE usually suffer from error sensitivity when irrelevant object images incorporated in texts. To address these issues, Chen, X., et al. propose a novel hierarchical visual prefix fusion NeTwork (HVPNeT) for visual-enhanced entity and relation extraction, aiming to achieve more effective and robust performance. Specifically, the paper regards visual representation as pluggable visual prefix to guide the textual representation for error insensitive forecasting decisions [106]. The paper further proposes a dynamic gated aggregation strategy to achieve hierarchical multi-scaled visual features as visual prefixes for fusion. Existing MNER methods are vulnerable to some implicit interactions and are prone to overlooking the involved significant features. To tackle this problem, X. Wang et al. proposed refining the cross-modal attention by identifying and highlighting some task-salient features [107]. The saliency of each feature is measured according to its correlation with the expanded entity label words derived from external knowledge bases. The paper further propose an end-to-end Transformer based MNER framework, which holds a neater architecture yet achieves better performance than previous methods.

*4.3. Summary*

There is still a great deal of research on multi-modal learning in audiovisual speech recognition, graphic emotion analysis, collaborative annotation, matching and classification, alignment, and other representation learning that we have not covered here. This study only introduces a portion of the relevant literature. Owing to the early start of the multi-modal named entity recognition task, progress has been made in multi-modal named entity recognition of text images and text speech. In terms of multi-modal relationship extraction, the multi-modal relationship extraction of most text images considers the entity image as the visual representation of its target entity. However, sometimes an image contains more than one entity but may contain multiple entities, which can be explored by learning multiple entities and their relationships in an image. At present, research directions of multi-modal naming entity recognition and relationship extraction focus on text and images. In the future, multi-modal IE, such as text and speech, text, and videos, can be explored.

Multi-modal representation learning, while keeping the modal-specific semantics intact, helps narrow the heterogeneity gap, and multi-modal entity linking technology can help align the cross-modal information of the same entity. The existing part of construction work mainly relies on the metadata of multi-media data, rather than its own visual or audio characteristics, which have considerable limitations. Therefore, combining database resources, such as text and vision, increasing noise processing power, expanding the scope of entity alignment and link prediction, and conducting entity relationship mining can help existing models achieve better performance when considering text and visual features comprehensively. At the same time, it can also provide a data foundation closer to the ground truth for higher-level intelligent applications and has considerable application potential in recommendation systems, information retrieval, visual questions and answering, and human-computer interaction.

## 5. Discussions

In summary, the characteristics of deep learning in entity relationship extraction, event extraction, and multi-modal information extraction are shown in Table 4:

**Table 4.** The characteristics of DL models in different IE tasks.

| Task | Classification | Pros | Cons |
| --- | --- | --- | --- |
| Entity relationship extraction | Supervised learning—pipeline extraction | Solves problems in stages and steps, with high flexibility of the model. | There are problems, such as error accumulation, ignoring the internal relationship and dependence between the two, and information redundancy; the problems of entity overlap, relationship overlap, and data noise cannot be solved. |
| | Supervised learning—joint learning | Makes full use of the relationship between entities and relationships to alleviate the problems of error accumulation and information redundancy. | The problem of entity overlap and relationship overlap cannot be solved. |
| | Distance supervised learning | Saves time and cost without a lot of manual labeling. | The problems of information noise and feature extraction error propagation need to be further solved. |

**Table 4.** *Cont.*

| Task | Classification | Pros | Cons |
| --- | --- | --- | --- |
| Event extraction | Single task | Solves problems in stages and steps, with high flexibility of the model. | Unable to get the relationship between events. |
| | Multitasking | Trigger words and argument information promote each other, improves the extraction effect, and effectively alleviates the problem of data sparsity. | The model structure design is complex and often needs to be completed with multiple models. |
| | Sentence level | High recognition accuracy. | Incomplete extraction of event information. |
| | Chapter level | Effectively extracts comprehensive information of events. | The recognition accuracy is low, and the information fusion needs to be strengthened. |
| Multi-modal information extraction | Multi-modal entity recognition | Improves the effect of named entity recognition for modal information, and multi-modal entity linking technology helps entity alignment. | Modal fusion needs to be improved, and the distinction between entities that are easy to be confused needs to be strengthened |
| | Multi-modal relation extraction | Reduces the loss of data information. | The information between different modes is repetitive and noisy. |

### 5.1. Method Level

Multi-model and multi-task joint information extraction. Compared with entity relation extraction, the amount of annotated data for event detection is relatively limited, and it is difficult to improve the event detection performance of a single task and a single model. The use of multi-model combined and multi-task learning to break through the bottleneck of the existing event recognition model has become an important research problem. It is of great significance to use multi-class models to represent sentence vectors, improve the attention mechanism, and constantly explore the differences in the effects of entity recognition and relationship extraction based on the mode of multi-task extraction models, such as hard sharing, soft sharing, and sharing-private extraction.

Information extraction based on knowledge enhancement. It has been demonstrated that knowledge embedding, knowledge enhancement, or knowledge distillation can significantly enhance the level of information extraction. Applying knowledge attention to specific tasks and trying to use remote supervision to apply prior knowledge or "teacher model" to enhance the model to capture the semantic relationship between entities and relationships has great research value for auxiliary information extraction. Exploring more useful external knowledge, such as prior knowledge to enhance the feature extraction and processing of noise problems, to improve the remote monitoring signal is also the direction of continuous exploration.

Multi-modal-based information fusion. In the context of the gradual evolution of artificial intelligence from a single mode to a multi-modal one and from perceptual intelligence to cognitive intelligence, the interaction between multi-modal data learning and knowledge graphs provides an extremely imaginative possibility for the landing of the value of big data in the application. With the expansion of the scope of the construction source data, a more comprehensive modal level, more fine-grained knowledge extraction, and richer semantic correlation will be the future development direction of the multi-modal

knowledge graph. In this context, the first problem is to explore the extraction effects of different modalities of information and make the model adaptively extract valuable content features from massive multi-modal data. Therefore, content feature extraction and multi-modal knowledge representation methods are constantly being explored by scholars.

*5.2. Model Level*

Model light weighting. At present, DL-IE research focuses on the optimization of network structure improvement. The larger the deep neural network model is, the more complex the structure, the more parameters, and the longer the model training time. How to reduce the model size while keeping the model performance unchanged is a direction for future research. Although deep learning models have been widely used in natural language processing or information extraction tasks, the storage and computation of models still face great challenges due to the demand for large amounts of data and the reliance on powerful computational resources. Therefore, it is a research trend of DL-IE to reduce the number of parameters and complexity of the model by compressing and accelerating the algorithm layer through model pruning, structural optimization design, knowledge distillation, quantization, and other methods while keeping the model performance unchanged.

Strengthening theoretical research. At present, the vast majority of researchers in DL-NLP focus their research on developing new models and optimal combinations of models, emphasizing experimental comparisons but lacking theoretical analysis and research, resulting in many deep learning models lacking a theoretical basis in natural language processing tasks and very slow performance improvements. Improving the maturity of the theoretical system of neural networks and exploring concise parameter forms and efficient training algorithms can surely bring more means of implementation and progress to knowledge extraction.

Improving model generalization ability. Deep neural network models are poorly interpretable and have made little progress in research on natural language generation tasks. Regardless of how large the database is, it cannot contain all the knowledge. Giving machines the ability to generalize outside the corpus of data for machine learning and giving them the ability to learn data outside the database efficiently and accurately is an inevitable product of deep learning knowledge extraction.

## 6. Prospect

The purpose of cognitive intelligence is to equip computers with human capabilities, such as knowledge representation, autonomous learning, and logical reasoning, and in the process, to make machines truly "rational" and able to explain the process and results of reasoning. The basis of cognitive intelligence is knowledge extraction and representation. Knowledge is not simply data or ordinary information but reflects the relationship between things in the objective world. For computers, how to extract the knowledge they need from information and big data in various formats on the internet is an important problem in knowledge extraction. Extracting domain-specific knowledge from unstructured multi-modal data becomes more and more important. Therefore, in the future, the study of unstructured, multi-modal, and multi-modal collaborative domain-specific knowledge acquisition will become the focus of further in-depth research by researchers.

**Author Contributions:** Conceptualization, Y.Y.; Data curation, F.G.; Formal analysis, Z.W. (Zhiwei Wang); Investigation, Z.W. (Zhilei Wu); Supervision, Y.Y. (Yuexiang Yang); Writing—original draft, S.L.; Writing—review & editing, Y.Y. and Z.W. (Zhilei Wu). All authors have read and agreed to the published version of the manuscript.

**Funding:** This research was funded by the National Key R&D Program of China (2021YFF0600400), the Key Project of Beijing Social Science Fund (19YJA001), the Special Funded Project for the Social Sciences of the Basic Scientific Research of the Central Universities (2021SKGL01), and the Special Funded Project for the Basic Scientific Research of the Central Universities for Postgraduates (2022YJSGL05).

**Institutional Review Board Statement:** Not applicable.

**Informed Consent Statement:** Not applicable.

**Data Availability Statement:** Not applicable.

**Conflicts of Interest:** The authors declare no conflict of interest.

## Appendix A. Summary of Characteristics and Applications of Typical Methods

| | Typical Model | Characteristic | Application |
|---|---|---|---|
| Entity relation extraction | RNN (2012) | Handles both internal feedback connections and feed-forward connections between the units. | Suitable for the extraction of timing features. |
| | CNN (2014) | The structure is simple, and the neural network is used for feature extraction, which avoids the tedious manual feature extraction of [13]. | Suitable to process data with correlation. |
| | PCNNs (2015) | Regard the remote-supervised relation extraction problem as a multi-instance problem, and a convolution structure with a segmented maximum pool is used to automatically learn related features. | Remote supervised relationship extraction. |
| | CAN (2018) | Produces deep semantic dependency features, introducing attention mechanisms to capture dependency representation [18]. | Chemistry-disease field. |
| | LSTM (2014) | Produces long-term dependencies from the corpus, but the network structure is more complex. | Machine translation, dialogue generation, encoding, and decoding. |
| | K-CNN (2019) | Contains two collaborative channels: the knowledge-oriented channel and data-oriented channel and combines the information obtained from the two channels [19]. | Causal relationship extraction. |
| | CRN (2020) | Extracts non-superordinate relationships from the unstructured text [20]. | Food field. |
| | 2ATT-BiGRU (2020) | Uses the character-level and sentence-level attention mechanisms, finds the words that have a significant impact on the output, and gives them a higher weight to better obtain their semantic information [25]. | Medical domain. |
| | FastText-BiGRU-Dual Attention (2021) | Focuses on words with decisive influence on sentence relationship extraction, creates word-level low-dimensional entity vectors at the embedding layer, and feeds the word embedding and position embedding results to the BiGRU layer to obtain high-level features [11]. | Forestry field. |
| | GREG (2020) | The two modules of the model are synchronized during training, and each of the model's modules is designed to deal with local relationships and global relationships separately [26]. | Overall relationship extraction. |
| | SKEoG (2021) | Takes full advantage of the document structure and external knowledge of [9]. | Medical relationship extraction at the document level. |
| | AGCN (2020) | Uses context and structural knowledge, combines GCN and a recurrent network-based encoder, and employs a new attention-based pruning strategy [30]. | Drug-drug interaction extraction. |
| | FC-GCN (2021) | Creates the un-directed graphs based on the combined features, uses the atomic features as the nodes, constructs the edges between the nodes according to the combination rules, and considers the prior knowledge and avoids the error caused by the resolution [31]. | Better analysis of the sentence structure. |
| | IPR-DCGAT (2021) | A dense connectivity graph attention network is used to update the representation of nodes and a two-step iterative algorithm to update the representation of edges [34]. | Document-level relationship extraction. |
| | BioGraphSAGE (2021) | Biological semantic and positional features are combined to improve the identification of long-distance entity relationships [35]. | Biological entity relationship extraction. |

| | Typical Model | Characteristic | Application |
|---|---|---|---|
| | KERE (2020) | Extracting knowledge information from the knowledge graph to generate knowledge-oriented word embeddings can enhance the effectiveness of the word embedding and using lexical features as supplementary information for semantic understanding can reduce semantic ambiguity and manually annotate [41]. | Biomedical relationship extraction. |
| | NAM (2019) | [43] demonstrates combining contextual information and knowledge representation with an attention mechanism. | Chemical-disease relation extraction. |
| | ATLOP (2021) | Adaptive thresholds replace global thresholds, and the local context pool shifts attention from the pre-trained language model to the localization-relevant context, alleviating multi-label and multi-entity problems [45]. | Document-level relationship extraction in biomedical fields. |
| | HDP (2019) | Converts the multi-pair-triplet extraction into a sequence generation task; generates a hybrid binary-pointer sequence extraction to alleviate the entity overlap problem [53]. | Multi-relationship triples were extracted. |
| | KSBERT (2021) | Integrates domain-specific external lexical and syntactic knowledge into end-to-end neural networks to solve the overlap problem [54]. | Military entities and relationship extraction. |
| | KEWE-BERT (2021) | Overlays the token embeddings and knowledge embeddings of BERT and TransR output with [55]. | Construction of the manufacturing domain knowledge map. |
| | Gated-K-BERT (2021) | Combines knowledge-aware attention representations with BERT; entities and their relations are jointly extracted using a gating fusion sharing mechanism [56]. | Study of the associations between depression and cannabis use. |
| | GraphRel (2019) | Combining the RNN and GCN, extracting the sequence features and region-dependent features of each word, considering the recessive features between word pairs, and considering the interaction between named entities and relationships through the relationship-weighted GCN [57]. | Joint entity and relationship extraction; alleviate overlap problems. |
| | A2DSRE (2021) | Advanced graph neural network GeniePath is introduced in DSRE to incorporate additional supervised information from the knowledge graph through the margin between the representation of the retraction package and the pre-trained knowledge graph embeddings [67]. | Reduce noise and remote supervision relationship extraction. |
| | REIF (2022) | Bridges the gap of realizing the influence of sub-sampling in deep learning. | Solve the problem of high noise interference. |
| Event extraction | DMCNN (2015) | Vocabulary and sentence-level features can be automatically extracted from plain text without complex NLP preprocessing; using dynamic multi-pooling layers to store more valuable information based on event triggers and event arguments [71]. | Single-task event extraction. |
| | JRNN (2016) | Based on the bi-directional RNN, introducing the memory matrix can effectively capture the dependencies between the argument element-roles and the trigger sub-types [73]. | Multi-event information extraction. |
| | DCFEE (2018) | Remote monitoring automatically tags event reference annotation triggers and arguments throughout the document, which includes sentence-level event extraction and document-level event extraction [87]. | Online event extraction of financial news and Chinese financial texts. |
| | HNN-EE (2019) | In the entity extraction module, the BiLSTM is used to capture the long-distance dependence information; in the event extraction module, the self-attention layer captures the internal structure of the sequence, and the gated convolution layer extracts the higher-level feature [86]. | Joint entity and event extraction. |
| | CNN-BiGRU (2021) | The word vector and position vectors are stitched as input, and word-level features are extracted using CNN and sentence-level features [85] using BiGRU. | Event-trigger word extraction. |
| | DEGREE (2022) | A data-efficient model that formulates event extraction as a conditional generation problem. | Focus on low-resource end-to-end event extraction. |
| | UIE (2022) | Structured extraction languages operate to uniformly encode different extraction structures, adaptively generate target draws, and capture the common IE capability [80] through large-scale pre-trained "text-to-structure" models. | Unified extraction of entities, relationships, and events in general fields. |

| | Typical Model | Characteristic | Application |
|---|---|---|---|
| Multimodal information extraction | IKRL (2017) | Considers the visual information in solid images and constructs representations for all images of entities using a neural image encoder, which are integrated into the aggregated image-based representations of [101] via an attention-based approach. | Triplet classification, knowledge map construction. |
| | Rp-BERT (2021) | Expands vanilla BERT to a multi-task framework of text-image relationship classification and visual-language learning; and better utilize visual information according to the relationship between text and image [97]. | Better use of visual information for multi-modal named entity recognition. |
| | UMGF (2021) | Uses unified multi-modal graphs and represents input sentences and images to capture various semantic relationships between multi-modal semantic units (words and visual objects); stacks multiple graph-based multi-modal fusion layers and iteratively performs semantic interactions to learn node representation [99]. | Explore the multi-modal graph neural networks of MNER for multi-modal named entity identification. |
| | MEGA (2021) | Generates scene maps from the image rich in visual information, treats the object features in the extracted scene map as visual semantic features, aligns the structure and semantic information of multi-modal features respectively, and then merges the alignment results [104]. | Social media relationship extraction. |
| | CAT-MNER (2022) | Proposed to refine the cross-modal attention by identifying and highlighting some task-salient features. The saliency of each feature is measured according to its correlation with the expanded entity label words derived from external knowledge bases. | To solve the problem that existing MNER methods are vulnerable to some implicit interactions and are prone to overlook the involved significant features. |
| | HVPNeT (2022) | Regards visual representation as a pluggable visual prefix to guide the textual representation for error-insensitive forecasting decisions. | To solve the problem that existing approaches for MNER and MRE usually suffer from error sensitivity when irrelevant object images incorporated in texts. |

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
