# Peer review of "A Survey of Information Extraction Based on Deep Learning"

_applsci, doi:10.3390/app12199691_

Round 1

Reviewer 1 Report

This paper was supposed to explain the basic concepts of deep learning and achievements of deep learning technology in the field of natural language processing from the three aspects of entity and relationship, event, and multimodal IE, and make a comparative analysis of various extraction techniques. The authors also summarized the prospects and development trends in deep learning in the field of natural language processing as well as difficulties requiring further study. However, I do not think this paper is good for publishing as a survey paper. In general, this paper does not meet the standard of a survey paper. My main concerns are as follows.

This paper did not provide a clear understanding/definition. The scope of this paper is not clear.

This paper planned to talk about Information Extraction Based on Deep Learning. However, it turns out to discuss more on Information Extraction.  In addition, lots of basic concepts of Information Extraction are discussed. This is not helpful and unnecessary at least for such a survey. There are a number of books that have very good introduction. What are the values by looking at this paper?

The Introduction Section should be rewritten. For instance, motivations for the survey and theoretical background of deep learning and Information Extraction should be provided.

The differences between Information Extraction Based on Deep Learning and Information Extraction Based on classical/machine learning methods should be highlighted from different perspectives.

It is recommended that authors should first describe the different types of Information Extraction methods and review their applications. Tables and schematic diagrams should be used to clarify the essential difference of different types of methods, and discuss their pros and cons.

Reviewer 2 Report

-Abstract should say more about the novelties of the surve

-All of the figures needs to be improved.

-Some further references concerning deep learning and convolutional neural networks should be added: 

-Gu, Jiuxiang, et al. "Recent advances in convolutional neural networks." Pattern recognition 77 (2018): 354-377.

-LeCun, Yann, Yoshua Bengio, and Geoffrey Hinton. "Deep learning." nature 521.7553 (2015): 436-444.

  -Dimitri, Giovanna Maria, et al. "Unsupervised stratification in neuroimaging through deep latent embeddings." 2020 42nd Annual International Conference of the IEEE Engineering in Medicine & Biology Society (EMBC). IEEE, 2020.

-Gu, Jiuxiang, et al. "An empirical study of language cnn for image captioning." Proceedings of the IEEE international conference on computer vision. 2017.

-Lavin, Andrew, and Scott Gray. "Fast algorithms for convolutional neural networks." Proceedings of the IEEE conference on computer vision and pattern recognition. 2016.

-All of the figures should be highly improved. They are too small and low quality. Make sure to make all of them readable, and add further descriptions to the captions 

-Prospect paragraph should be named discussion

-Novelty of the proposed survey structure should be highlighted. What is new in this survey with respect to previous ones?

-English should be improved, please read carefully and improve

Reviewer 3 Report

The work presented in this paper discusses Information Extraction Based on Deep Learning. I would suggest the following suggestions to further improve the quality of the manuscript.

1. Abstract can be further improved by highlighting the following points:

1.a: What is the rationale behind information Extraction Based on Deep Learning, what is its primary purpose?

1.b: What are the main objectives for using deep learning-based techniques.

1.c: What parameters you have considered for comparing the works?

Reorganize the abstract to conclude:

(a) The overall purpose of the study and the research problems you investigated.

(b) The basic design of the study.

(c) Major findings or trends found as a result of the study.

(d) A brief summary of your interpretations and conclusions.

2. The introduction needs to clarify the (1) motivation, (2) challenges, (3) contribution, (4) objectives, and (5) significance/implication. All the information (should be) presented in sequence idea.

3. How do the extraction and joint extraction differ?

4. Compare different types of extraction using a table by considering various measures.

5. Analysis of various existing works should be elaborated.

6. Classify different techniques based on their extraction ability.

7. Add more recent references to enhance the literature survey section.

8. Any comparative analysis to testify that this study is more advanced than others? 

Round 2

Reviewer 1 Report

The contribution of the manuscript is still very limited and it is not satisfactory as a survey paper.

Reviewer 2 Report

The authors have addressed my comments. However still some important aspects are missing: 

1) Can you make a table with the comparisons, dates of the various methodologies?

2) All of the figures need to be highly improved again. Captions need to be extended, and figures enlarged. Figure 1 is unreadable, and so are almost all of the others 

3) Some more deep learning important references should be added to the paper, to make it more comprehensively reviewing the field of interest. As for example: 

   -Dimitri, Giovanna Maria, et al. "Unsupervised stratification in neuroimaging through deep latent embeddings." 2020 42nd Annual International Conference of the IEEE Engineering in Medicine & Biology Society (EMBC). IEEE, 2020.

 -LeCun, Yann, Yoshua Bengio, and Geoffrey Hinton. "Deep learning." nature 521.7553 (2015): 436-444

-Minaee, Shervin, et al. "Image segmentation using deep learning: A survey." IEEE transactions on pattern analysis and machine intelligence (2021)

-Garcia-Garcia, Alberto, et al. "A survey on deep learning techniques for image and video semantic segmentation." Applied Soft Computing 70 (2018): 41-65.

-Wang, Panqu, et al. "Understanding convolution for semantic segmentation." 2018 IEEE winter conference on applications of computer vision (WACV). Ieee, 2018.

-Alalwan, Nasser, et al. "Efficient 3D deep learning model for medical image semantic segmentation." Alexandria Engineering Journal 60.1 (2021): 1231-1239

-Bianchini, Monica, et al. "Deep neural networks for structured data." Computational Intelligence for Pattern Recognition. Springer, Cham, 2018. 29-51

4) Rename the last paragraph. Instead of prospect call it conclusions and draw the summary of the paper and the future perspective of the paper 

Reviewer 3 Report

I am now satisfied with the revision.

Author Response

Thanks very much !

Round 3

Reviewer 1 Report

Accept

Author Response

Thank you for your guidance.